



# Impact of a subtropical high and a typhoon on a severe ozone pollution episode in the Pearl River Delta, China

Shanshan Ouyang[1,2], Tao Deng[2], Run Liu[1,3], Jingyang Chen[4], Guowen He[2,5], Jeremy Cheuk-Hin Leung[2], Nan Wang[2], Shaw Chen Liu[1,3]

[1]Institute for Environmental and Climate Research, Jinan University, Guangzhou, 511443, China
[2]Institute of Tropical and Marine Meteorology/Guangdong Provincial Key Laboratory of Regional Numerical Weather Prediction, China Meteorological Administration, Guangzhou 510640, China
[3]Guangdong-Hongkong-Macau Joint Laboratory of Collaborative Innovation for Environmental Quality, Guangzhou, 511443, China
[4]Guangdong Ecological Meteorology Center (Pearl River Delta Center for Environmental Meteorology Prediction and Warning), Guangzhou 510640, China
[5]School of Atmospheric Sciences, Sun Yat-sen University, Zhuhai 519082, China

*Correspondence to*: Tao Deng (tdeng@gd121.cn) and Shaw Chen Liu (shawliu@jnu.edu.cn)

**Abstract.** A record-breaking severe ozone ($O_3$) pollution episode occurred in the Pearl River Delta (PRD) in early Autumn 2019 when PRD was under the influence of a Pacific subtropical high followed by Typhoon Mina. In this study, we analyzed the effects of meteorological and photochemical processes on the $O_3$ concentration in PRD during this episode by carrying out the Weather Research Forecast-Community Multiscale Air Quality (WRF-CMAQ) model simulations. Results showed that low relative humidity, high boundary layer height, northerly surface winds and strong downdrafts were the main meteorological factors contributing to $O_3$ pollution. Moreover, delayed sea breezes that lasted into the night would transport $O_3$ from the sea back to land and resulted in secondary $O_3$ maxima at night. In addition, $O_3$ and its precursors stored in the residual layer above the surface layer at night can be mixed down to the surface in the next morning, further enhancing the daytime ground-level $O_3$ concentration the following day. Photochemical production of $O_3$, with daytime average production rate of about 7.2 ppb/h, is found to be the predominate positive contributor to the $O_3$ budget of the boundary layer (0-1260m) during the entire $O_3$ episode; while the horizontal and vertical transport fluxes are the dominant negative contributors. This $O_3$ episode accounted for 10 out of the yearly total of 51 days when the maximum daily 8-h average (MDA8) $O_3$ concentrations exceeded the national standard of 75 ppb in PRD in 2019. Based on these results, we propose that the enhanced photochemical production of $O_3$ during the episode is a major cause of the most severe $O_3$ pollution year since the official $O_3$ observation started in PRD in 2006. Moreover, since this $O_3$ episode is a synoptic scale phenomenon covering the entire eastern China, we also suggest that the enhanced photochemical production of $O_3$ in this $O_3$ episode is a major cause of the extraordinary high $O_3$ concentrations observed in eastern China in 2019.



# 1 Introduction

Tropospheric ozone ($O_3$) is a product of photochemical reactions between volatile organic compounds (VOCs) and nitrogen oxides ($NO_x$) under the sunlight; it is a typical secondary pollutant that plays a major role in regional atmospheric pollution (Sillman. 1999; Trainer et al., 2000; Lu et al., 2018). High surface $O_3$ concentrations have adverse effects on human health (Jacob and Winner, 2009; Fleming et al., 2018; Liu et al., 2018) and crops production (Wang et al., 2017; Mills et al., 2018). In the past few decades, along with the rapid economic development, air pollution problems have become increasingly serious in China. Since the implementation of air pollution prevention and control measures in 2013, the overall air quality in China, particularly the concentration of particulate matter, has been significantly improved (Zhang and Geng, 2019). However, in recent years the summer and autumn $O_3$ concentration in eastern China, particularly in Beijing-Tianjin-Hebei (Gong and Liao, 2019; Mao et al., 2020), Yangtze River Delta (Shu et al., 2016; Zhan et al., 2020), and Pearl River Delta (PRD) (Deng et al., 2019; He G et al., 2021) actually increasingly exceeded China's national ambient air quality standards (hourly $O_3$ of 200 $\mu g/m^3$, and the maximum daily 8-h average (MDA8) $O_3$ concentrations of 160 $\mu g/m^3$).

Variations in the emission of $O_3$ precursors and meteorological conditions are two main factors affecting the atmospheric $O_3$ concentration (Xu et al., 2018; Han et al., 2019). $NO_x$ produced by industry, transportation, and power plants, and VOCs from solvent use, industry, transportation, residential and vegetation are major sources of the $O_3$ precursors (Li et al., 2017; Zheng et al., 2018). Meteorological conditions such as high temperature, low humidity, high pressure, low wind speed, and strong solar radiation can affect the photochemical production and transport of $O_3$, resulting in high $O_3$ pollution events (Deng et al., 2019; He C et al., 2021; Hu et al., 2021). Located in the coastal area of South China, PRD region has a typical subtropical monsoon climate, in which the weather conditions are easily affected by typhoons and subtropical highs in summer and autumn (Lin et al., 2019). Since these two synoptic meteorological patterns are highly conducive to $O_3$ generation, the study of $O_3$ generation under these conditions are essential to the understanding of $O_3$ pollution problem in PRD.

Previous studies have reported the influence of the intensity (Lam et al., 2018), track (Deng et al., 2019), and occurring frequency (Lin et al., 2019) of tropical cyclones on the $O_3$ concentration in PRD. Furthermore, by comparing the meteorological conditions and the $O_3$ sources in summer and autumn with and without typhoons, Qu et al. (2021) revealed that the approach of typhoons accompanied by higher wind speeds and strengthened downdrafts would reduce cloud cover and thus led to higher solar radiation, which was favorable to the $O_3$ production. Zhan et al. (2020) analyzed $O_3$ production processes caused by four consecutive typhoons in the summer of 2018 based on model simulations. They found that $O_3$ pollution events in the YRD region mainly occurred between the end of a typhoon and the arrival of the next typhoon. Since fluctuations between high-pressure and low-pressure systems strongly affect the variations in $O_3$ concentrations (Fiore et al., 2015), the Western Pacific Subtropical High (WPSH) is also an important factor affecting $O_3$ in eastern China (Zhao and Wang, 2017; Chang et al., 2019; Yin et al., 2019). Numerical simulation studies by Zeren et al. (2019) and Shu et al.



(2016) revealed that strong photochemical reactions and unfavorable diffusion conditions caused by the single/combined action of subtropical highs and typhoons are the main reasons for the occurrence of regional $O_3$ pollution.

In this study, we carry out Weather Research Forecast-Community Multiscale Air Quality (WRF-CMAQ) model simulations and make comprehensive analyses of meteorological and photochemical processes in a severe $O_3$ pollution episode associated with a Pacific subtropical high and Typhoon Mina in 2019. The rest of this paper is structured as follows. Data and methods are presented in Section 2. Section 3 contains the major results and findings. It is subdivided into five subsections, namely, basic characteristics of the regional $O_3$ episode, evaluation of model performance, influence of

meteorological conditions on $O_3$ during the three periods, characteristics of $O_3$ in the horizontal and vertical spatial distribution, and contributions of photochemical and transport processes to $O_3$ formation. A summary and conclusions are presented in Section 5.

## 2 Data and methods

### 2.1. Datasets

Hourly $O_3$ concentration monitoring data during the $O_3$ pollution episode in PRD from September 1 to October 31, 2019 were obtained from the China National Environmental Monitoring Center (NEMC) (available at https://quotsoft.net/air/). The study area includes 56 stations in nine cities (Zhaoqing (ZQ), Jiangmen (JM), Foshan (FS), Zhuhai (ZH), Zhongshan (ZS), Guangzhou (GZ), Dongguan (DG), Shenzhen (SZ), and Huizhou (HZ)) in PRD (Figure 1b).

The European Centre for Medium-Range Weather Forecasts (ECMWF) Reanalysis v5 (ERA5) dataset (available at

80 https://cds.climate.copernicus.eu/) with a horizontal resolution of 0.25° × 0.25° and a time interval of 6 hours were used to analyze the atmospheric circulation patterns during the pollution episode. The variables used in this study include zonal, meridional wind speeds and geopotential height.

Hourly meteorological data of PRD region were provided by the Guangdong Meteorological Service, including 2-m temperature ($T_2$), 2-m relative humidity (RH), 10-m wind speed ($WS_{10}$), and 10-m wind direction ($WD_{10}$).

In this study, the 6-hour Final Global Forecast System Operational Analysis (FNL) data with a resolution of 0.25°×0.25° from the National Center for Environmental Prediction (NCEP) were used to provide initial and boundary conditions for the WRF simulation (https://rda.ucar.edu/datasets/). Geographical data were obtained from the Research Data Archive of the National Center for Atmospheric Research (NCAR) (http://www2.mmm.ucar.edu/wrf/users/downloads. html).

### 2.2. Model description and configurations

The WRF (v3.9.1)-CMAQ (v4.7.1) model, which has been extensively shown to perform well simulating pollution processes in China (Wang et al., 2015; Zhan et al., 2020; Qu et al., 2021; Zhao et al., 2021), was used to simulate the $O_3$ pollution episode. The WRF model was set with two one-way nested domains with horizontal resolutions of 27 km and 9 km, respectively (Figure 1a). The outer domain (d01) covers most areas of East Asia with 283 × 184 grids, and the inner domain



(d02) covers most parts of South China with 250 × 190 grids. For both domains, there were 38 vertical sigma layers extending from the surface to the top pressure of 50 hPa, with 17 layers located below 1 km providing vertical information on the planetary boundary layer. The model applies the Rapid Radiative Transfer Model (RRTM) longwave scheme, Mesoscale Model (MM5) similarity surface layer, Noah land surface model, ACM2 planetary boundary layer scheme, and Grell-Devenyi (GD) ensemble scheme. The detailed configuration options for dynamic parameterization in WRF are summarized in Table 1. The resolutions of CMAQ were 182 × 138 and 220 × 170 grids. Biogenic emissions were generated offline using the Model of Emissions of Gases and Aerosols from Nature (version 2.04) (Guenther et al., 2006). The chemical mechanism of carbon bond 05 (Yarwood et al., 2005) was chosen for gas-phase chemistry. The anthropocentric pollutant emissions were obtained from the Multi-resolution Emission Inventory for China (MEIC) of 2016 (http://meicmodel.org/). The period simulated in this study was from 00:00 (Local Time, LT) on September 18 to 00:00 (LT) on October 5, in which the first 72 h were taken as the spin-up time to minimize the bias due to initial conditions.

Integrated process rate (IPR) is an effective diagnostic module provided by the CMAQ model that allows the calculation of the hourly contribution of different physicochemical processes to various pollutants, thus determining the quantitative impact of each process on the change in pollutant concentration in each grid cell. This method makes it possible to identify the causes of pollution and the main physicochemical processes responsible for changes in pollutant concentrations. The causes of pollutant concentration variations were classified into seven types of physical and chemical processes: horizontal advection (HADV), vertical advection (ZADV), horizontal diffusion (HDIF), vertical diffusion (VDIF), dry deposition (DDEP), cloud processes (CLDS), and chemical processes (CHEM). In this study, the horizontal transport (HTRA) was defined as the sum of HADV and HDIF, and the vertical transport (VTRA) was defined as the sum of ZADV and VDIF.

### 2.3. Model evaluation

To evaluate the model performance, the simulation results in d02, including $T_2$, RH, $WS_{10}$, $WD_{10}$, and $O_3$ concentrations, were compared with hourly observation data from NAWO and CNMC. Statistical metrics including the correlation coefficient $R$, root-mean-square error (RMSE), normalized mean bias (NMB), and index of agreement (IOA) (Huang et al., 2005) were used. These metrics are defined as follows:

$$R = \frac{\sum_{i=1}^{N}(S_i-\bar{S})(O_i-\bar{O})}{\sqrt{\sum_{i=1}^{N}(S_i-\bar{S})^2}\sqrt{\sum_{i=1}^{N}(O_i-\bar{O})^2}}, \tag{1}$$

$$\text{RMSE} = \sqrt{\frac{\sum_{i=1}^{N}(S_i-O_i)^2}{N}}, \tag{2}$$

$$\text{NMB} = \frac{\sum_{i=1}^{N}(S_i-O_i)}{\sum_{i=1}^{N}O_i} \times 100\%, \tag{3}$$

$$\text{IOA} = 1 - \frac{\sum_{i=1}^{N}(S_i-O_i)^2}{\sum_{i=1}^{N}(|S_i-\bar{O}|+|O_i-\bar{O}|)^2}, \tag{4}$$

where $S_i$ and $O_i$ represent the simulations and observations, respectively; $\bar{S}$ and $\bar{O}$ represent the mean of the simulated and observed values, respectively; and $N$ is the number of valid data. $R$ represents the degree of matching between the





observation data and simulation data, and NMB and RMSE indicate the degree of deviation between the observation and
simulation data. In general, when the NMB and RMSE are closer to zero, the $R$ and IOA are closer to one, the model
simulation is considered to be in better agreement with the observation.

# 3 Results and discussions

## 3.1. Characteristics of the $O_3$ episode

### 3.1.1 Overview of the $O_3$ episode

Figure 2a shows the diurnal variation of $O_3$ concentration observed in PRD region (averaged over the 56 stations) from
September 1 to October 31, 2019. It is worth noting that the subtropical highs and typhoons were the two main synoptic
systems when the $O_3$ episode occurred (the yellow and blue shades). In particular, $O_3$ concentrations persistently exceeded
the national ambient air quality secondary standard of about 93 ppb (the red dotted line) in the afternoons from September 25
to October 2 for 8 straight days. In addition, there were 51 days when MDA8 $O_3$ exceeded the secondary standard of about
75 ppb in PRD in the entire year of 2019 (not shown). Figure 2b further shows the calendar chart of MDA8 $O_3$
concentrations in nine cities from September to October, sorted by longitude in PRD. The $O_3$ concentration in PRD remained
at a relatively high level in late September and early October, with MDA8 $O_3$ reaching Lev 3 in nine cities from September
25 to October 1, and even reaching Lev 5 in JM, ZS, and ZH on September 28.

In general, nighttime $O_3$ concentrations are low due to the titration by nitrogen oxide emitted during the night. However,
during the pollution period, the $O_3$ concentration rebounded at several sites after sunset, and the time for the rebound of $O_3$ at
different sites showed different time lags from coastal to inland sites. For example, on September 26, the times of $O_3$
rebound in SZ, DG, GZ, and FS were 20:00, 21:00, 22:00, and 23:00, respectively (Figure 3a). On September 29, the time of
$O_3$ rebound in SZ, ZH, ZS, and DG were 18:00, 20:00, 22:00, and 23:00, respectively (Figure 3b). On October 1, the times of
$O_3$ rebound in JM, FS, and GZ were 21:00, 22:00, and 23:00, respectively (Figure 3c). This phenomenon apparently was
related to the backflow of $O_3$ from the ocean due to sea breezes, which will be further elaborated in Section 3.4.2.

### 3.1.2. Evolution of synoptic systems

Figure 4 shows the spatial distribution of the ERA5 reanalysis 500 hPa geopotential height and 850 hPa wind field over East
Asia at 14:00 on September 25 and from September 29 to October 3. On September 21 (not shown), Typhoon Taba moved
northward away from PRD, and PRD region was controlled by low-level northerly airflow from the west side of the typhoon.
$O_3$ concentration dropped slightly at this time (Figure 2). From September 25 (Figure 4a) to 28, the area enclosed by the
5880 gpm isoline (orange area) continued to cover the entire PRD, which means the downdraft caused by the subtropical
high could suppress the vertical diffusion of surface air pollutants. Meanwhile, solar radiation intensified under the clear sky,
and caused $O_3$ to continue increase (Figure 2).





Afterward the position and intensity of the subtropical high was affected by Tropical storm Mina, which developed and
strengthened rapidly over the western Pacific on September 28 and was upgraded to typhoon level on September 29 (Figure
4b). It crossed the ridge of high pressure all the way northward to the southeast of Taiwan, resulting in a break in the
subtropical system. When the western extension of the ridge of the eastern subtropical high retreated eastward to 118°E,
PRD region was in the downdraft area outside the typhoon system. Meanwhile, high $O_3$ levels were observed continuously in
PRD. On October 1 (Figure 4d), Typhoon Mina made landfall on the coast of eastern China, then turned northeastward and
made landfall again on the coast of Korea on October 2 (Figure 4e). In the end, it merged into the upper trough of the
westerlies on October 3 (Figure 4f). As Typhoon Mina moved away, the ground-level $O_3$ concentration in PRD decreased
significantly.

In summary, PRD was mainly influenced by the WPSH followed by Typhoon Mina during the $O_3$ episode of September 23–
October 2. Therefore, the WRF-CMAQ model simulation and subsequent analyses will focus on three periods: the
subtropical high period (September 23–28), typhoon Mina period (September 29–October 2), and the clean period (October
3–4).

**3.2. Evaluation of model performance**

Hourly observations of $T_2$, RH, $WS_{10}$, and $WD_{10}$ at meteorological stations in nine cities in PRD from September 21 to
October 5, 2019 are compared with the WRF simulation results to evaluate the model performance (Figure 5). The results of
the evaluation metrics, $R$, NMB, RMSE, and IOA are listed in Table 2. Simulated $T_2$ and RH are consistent with the
observations, with $R$ of 0.97 and 0.84, respectively. WRF underestimates $T_2$ and RH by 1.92% and 0.97%, respectively, and
RMSE are 1.05 °C and 9.10%, respectively. Surface winds are closely related to the horizontal transportation, accumulation,
and diffusion of pollutants. Although WRF overestimates $WS_{10}$ in this study by 69.2%, $R$ value reaches 0.69, indicating that
the model can reproduce the variability of wind speed. The simulation of wind fields is influenced by the terrain and various
complex physical processes (Wang et al., 2015); however, the IOA value of $WD_{10}$ is 0.64, indicating that the model can
simulate well the variability in $WD_{10}$ during the study period. In general, the statistical metrics above show that WRF can
capture the main meteorological characteristics of this $O_3$ episode, similar to those of previous studies on $O_3$ episodes in
PRD (Wang et al., 2015; Li Y et al., 2021).

Figure 5 also shows the time series of $O_3$ in the observations and simulations. The model captures the diurnal variation of $O_3$
well, reaching the peak in the afternoon, then gradually decreasing to the low values at night. Although the $O_3$ concentrations
simulated by the CMAQ model are lower at nights of September 21 and 22 and higher in the afternoon of October 1–3, the
NMB of -14.25% and RMSE of 20.36 ppb indicate that the model results are within the acceptable ranges. The bias of the
model may come from the WRF simulation error and/or the uncertainty in emissions (Wu et al., 2021). The emission
inventory used is based on the MEIC prepared in 2016, which may not accurately represent real emissions in 2019. In
addition, the uncertainty of emissions of $O_3$ precursors ($NO_x$ and VOCs) may also lead to a negative bias in nighttime $O_3$ due
to the titration effect (Lu et al., 2019; Yang et al., 2019).





### 3.3. Influence of meteorological conditions on O$_3$ during the three periods

The effects of different weather systems on O$_3$ during the three periods of this O$_3$ episode are analyzed by examining the meteorological variables in Figure 6: T$_2$, RH, U$_{10}$, V$_{10}$, WS$_{10}$, planetary boundary layer height (PBLH), downward short-

wave flux at the ground surface (SWDOWN) and vertical velocity (Omega). The parameters closely related to photochemical processes, namely SWDOWN, T$_2$, RH and PBLH, are shown in daytime averages (08:00–18:00), while the parameters closely related to transport, namely U$_{10}$, V$_{10}$, WS$_{10}$, and Omega, are averaged over the entire 24 hours. To explore the vertical air motions below the boundary layer (1260 m), Omega was calculated from the average of all model layers below 1000 m. Figure 6a shows the values of different meteorological parameters selected for the subtropical high period

(September 25–28), typhoon period (September 29 to October 2), and clean period (October 3–4). SWDOWN shows nearly the same value in all three periods. Compared to the clean period, lower RH, higher PBLH, predominantly northerly winds at the surface (negative V$_{10}$) and stronger downdraft (positive Omega) were found in the first two periods. When PRD was under the influenced of Typhoon Mina, it had a higher T$_2$, enhanced northerly wind, higher WS$_{10}$ and stronger Omega compared to the subtropical high period.

The key meteorological parameters affecting the changes in O$_3$ concentration in PRD varied as Typhoon Mina moved away from PRD can be seen in Figure 6b: the temperature increased slightly, the northerly wind strengthened at first and then gradually weakened, the average WS$_{10}$ changed very little, while the downdraft decreased abruptly on October 1. Although Typhoon Mina gradually moved away from PRD on October 2, the subtropical high strengthened (Figure 3e), resulting in PRD being still under the control of strong subsidence airflow. On October 3, Typhoon Mina moved further away with the

weakening of the subtropical high, and PRD was located between the broken subtropical high (Figure 3f). With southerly winds prevailing at lower levels (positive V$_{10}$), the O$_3$ episode was greatly alleviated, indicating that the clean sea breeze in the removal of O$_3$ concentration played an important role.

To compare the height of the atmospheric mixed layer at different period, Figure 6c shows the virtual potential temperature profile of PRD from September 25–26 and 29 to October 4, at 14:00. A superadiabatic layer appeared at about 100 m near

the ground, the virtual potential temperature decreased with height. Comparing the heights at which the inflection point of $\theta_v$ occurs in different period, the boundary layer was higher at the subtropical high and typhoon periods (both above 1500 m). Furthermore, the $\theta_v$ were higher in the typhoon and clean period, but the mixed layer height was significantly lower in the clean period than before, indicating that an appropriately high mixed layer height allowed for downward transport of O$_3$ from the upper levels would be conducive to the development of near-surface pollution. The findings above suggest that the

meteorological factors contributing to this O$_3$ pollution are mainly low RH, prevailing northerly winds, strong downdrafts and high PBLH.



### 3.4. Horizontal and vertical spatial distributions of O₃

#### 3.4.1. Effect of prevailing wind on O₃

The horizontal spatial distributions of $O_3$ and wind fields at 16:00 on September 26, 30 and October 3 are selected to analyze
the impact of surface winds on $O_3$ in the three periods. Northeasterly winds prevailed during the subtropical high period
(Figure 7a), while north-northwesterly winds prevailed during the typhoon period (Figure7b). During both periods the $O_3$
concentration in the southern part of PRD was higher than that in the northern part, indicating that the northerly component
of winds delivered the high concentrations of $O_3$ in the northern PRD southward to the coastal areas and overseas. After
Typhoon Mina moved northward and dissipated on October 3, the winds in PRD shifted to southerlies (Figure 7c). At this
time, the southerly winds from the sea had a cleansing effect on the $O_3$ pollution. Under the transport of the southerly wind,
the $O_3$ concentration in the downwind northern part of PRD became higher than the upwind southern part. This further
verifies the important influence of southerly winds on the distribution of $O_3$ as described in Section 3.3.

#### 3.4.2. Effect of sea-land breeze

As mentioned earlier, an $O_3$ secondary peak was observed at several stations during the period under the influence of a
subtropical high or typhoon, and the time of the secondary peak at these stations was delayed from coastal to inland (Figure
3), indicating that the secondary peak was influenced by the circulation of the sea-land breeze. Many studies have shown that
sea-land breeze plays an important role in the transport of air pollutants between land and sea, and the interaction of sea-land
air masses would lead to the redistribution of $O_3$ in the coastal areas such as PRD (Ding et al., 2004; Wu et al., 2013; Wang
et al., 2018; Zeren et al., 2019; Lin et al., 2021). In order to explain the mechanism influencing sea-land breeze, Figure 8
shows the vertical distribution of $O_3$ concentration (contours) and atmospheric circulation (wind vectors) over PRD along the
WD10 at 14:00, 20:00 and 23:00 on September 26, 29 and October 1. At 14:00, the locally generated high concentration of
$O_3$ covered most of the land area of PRD, and a weaker sea breeze began to appear at the junction of sea and land in the near-
surface layer. Under the domination of the prevailing northerly wind, the high concentration of $O_3$ was gradually transported
to the coastal areas at 20:00, after which the coastal low and middle-layer $O_3$ was brought back to the land as the sea breeze
strengthened. The influence of the sea breeze could reach inland areas as far as FS and the impact height can reach more than
500 m. There is a clear stationary zone between the sea breeze and the northerly wind above the top of the boundary layer,
which is more conducive to the formation of the residual layer. Our analysis shows that the contribution of horizontal
transport to the increase in $O_3$ concentration during the influence of sea breeze was about 6.9 ppb on average, and up to 8.4
245   ppb on September 26.
It is well known that sea-land breeze emerges due to the surface temperature difference between the land and the ocean. The
land warms faster than the ocean during the day because the latter has a higher heat capacity than the former. As a result, the
sea breeze usually starts in the mid-morning when the land temperature gets higher than that of the sea. At night the land





cools faster than the ocean, triggering the land breeze when the land temperature becomes lower than the ocean temperature.
However, the sea breeze was still noticeable as late as 23:00 during this episode. The reason for this phenomenon is that under the influence of prevailing northerly winds, the occurrence of the sea breeze was delayed until approximately 14:00 (Wu et al., 2013; Wang et al., 2018). In addition, the existence of the heat island effect caused the temperature in the area near the Pearl River Estuary to remain higher than the ocean temperature until around 20:00 at night (Li et al., 2016; Zhan and Xie, 2022), after which it shifted to land breeze as the land temperature fell below that of the ocean.

These results indicate that when the land breeze direction coincides with the northerly background wind, the locally generated high-concentration $O_3$ is transported to the southern part of PRD and coastal sea (Figure 7). When the sea breeze in the opposite direction of the prevailing northerly wind appeared after sunset, the high-concentration $O_3$ transported to the sea was brought back to coastal areas or even inland, causing ground-level $O_3$ concentrations to have a secondary peak at night.

### 3.4.3. Effect of nighttime residual $O_3$ above the surface layer

The vertical motion of airflow, especially the subsidence airflow due to the subtropical high and typhoon, can have important impact on the $O_3$ pollution in PRD. Therefore, this section explores the vertical variation in $O_3$. The average $O_3$ concentration simulated by the model for the area (112.43–114.53° E, 22.23–23.39° N) was used to analyze the formation of $O_3$ during the $O_3$ episode. Figure 9 shows the vertical distribution of the $O_3$ concentration during the $O_3$ episode. After sunset, significant amount of $O_3$ could be seen stored in the nighttime residual layer (500 m–1000 m) as well as above the mixed layer (1000 m) during the $O_3$ episode. Take September 29 as an example, the ground-level $O_3$ concentration stayed above 100 ppb from noon until 16:00 in the afternoon. After sunset (around 18:00), the ground temperature dropped rapidly, and an inversion formed with the warm air in the upper boundary layer. Because of the absence of photochemical production and consumption of near-surface $O_3$ by NO titration, near-surface $O_3$ concentrations dropped sharply. Above the inversion layer, high daytime $O_3$ is stored because there was no titration of $O_3$ by NO. Higher $O_3$ of approximately 45–85 ppb accumulated in the residual and mixed layer around 500–1500 m at night.

High concentrations of $O_3$ in the residual layer above the surface layer were slowly transported to the surface until sunrise because the nighttime inversion slowed down the vertical mixing (yellow bars in Figures 10d-f). After sunrise, the contribution of photochemical production (CHEM) within the boundary layer began to increase (blue bars in Figure 10a-c); the height of the mixed layer kept rising and the boundary between it and the residual layer disappeared due to the development of the mixed layer. The contributions of vertical transport (VTRA, yellow bars in Figure 10a-c) were small during this time, indicating that the $O_3$ inflow from the upper layer was almost equal to the $O_3$ transport to the surface. Therefore, for the near-surface $O_3$, the contributions of VTRA may contain a significant contribution of CHEM in the layer above the surface layer. Quantitatively it is difficult to evaluate this contribution by CHEM, hence VTRA has to be treated as the maximum contribution from the residual layer.





The near-surface VTRA (yellow bars in Figure 10d-f) is substantially higher than CHEM (blue bars in Figure 10d-f), indicating that the largest increase in ground-level $O_3$ concentration the next morning is mainly due to the vertical mixing of higher concentrations of $O_3$ in the residual layer with near-surface air masses. As clean southerly winds prevailed in PRD on October 3, the updrafts in the boundary layer gradually increased, hence the contribution to $O_3$ from vertical mixing of the

residual layer and photochemical reactions diminished. Thus, even with higher $O_3$ storage in the residual and mixed layers, $O_3$ pollution was mitigated. The results of the process analysis show that VTRA (including contribution by CHEM above the surface layer) contributes 34%-50% to surface $O_3$ at 8:00-14:00 during the entire episode. This range of VTRA is consistent with those obtained by Li X et al. (2021) in eastern China (about 12.6%-78.3%), Zhu et al. (2020) in rural areas of the North China Plain (about 50.7%) and He et al. (2021b) in Shenzhen (about 47.44%-61.44%); but higher than Liu et al. (2022) in

urban areas of the North China Plain (about 20.6%-27.9%).

The above results suggest that high $O_3$ concentrations in a region can be generated by daytime photochemical reactions and that $O_3$ and its precursors stored in the residual layer during the night are mixed at the surface by vertical movement in the early morning, significantly impacting daytime ground-level $O_3$ the next day.

### 3.5. Contributions of photochemical and transport processes to $O_3$ formation

Figure 11a shows the vertical distribution of the daytime (08:00–18:00) contribution of individual photochemical and transport processes to the $O_3$ concentration in PRD in selected representative days of the subtropical high period (September 26), typhoon period (September 29), and clean period (October 3). As expected, the photochemical production of $O_3$ is the main positive contributor to the $O_3$ budget of all layers between 35m to about 1000m. The CHEM is balanced by the VTRA (the main negative contributor) to both the surface layer and layers above 1000m (the free troposphere). The surface layer $O_3$

is mainly maintained by the balance between VTRA and dry deposition (DDEP). Relative to the clean period, the CHEM is significantly greater and the vertical mixing was more intense when PRD is affected by the subtropical high or Typhoon Mina. Between the two pollution periods, the typhoon period has slightly greater CHEM and less horizontal transport/dispersion (HTRA) of $O_3$ than the subtropical high period.

The time series at hourly resolution of individual photochemical and transport processes in the boundary layer (defined as 0-

1260 m based on the median height of PBLH in Figure 6a) for the entire $O_3$ episode are shown in Figure 11b. CHEM dominates the positive contribution to $O_3$ during the day from about 08:00 to 15:00 (exact time is more clearly illustrated in Figures 10a, 10b and 10c). This is the case throughout the entire episode, even for September 24, October 3 and 4, indicating clearly that the daily $O_3$ maximum is primarily controlled by CHEM. The horizontal and vertical transport flux (HTRA+VTRA) is the dominant negative contributor to the $O_3$ formation throughout the whole day. The absolute value of

HTRA+VTRA starts increasing in the morning, reaching a peak value near the $O_3$ maximum where it overtakes CHEM and causes $O_3$ concentration to decrease in the afternoon and thereafter. This near-balance between CHEM and HTRA+VTRA at the daily $O_3$ maximum occurred throughout the entire $O_3$ episode, even for the clean period. The results above have an important implication: the photochemical production of $O_3$ in the boundary layer in PRD during the $O_3$ episode contributes



to, not only the high O₃ inside the boundary layer in PRD, but also the transport of O₃ horizontally and vertically outside the
boundary layer in PRD.

Given the importance of the daytime CHEM, the daytime average CHEM for each period is shown in Table 3 and compared
to values of VTRA, HTRA, DDEP and O₃. Again, the dominant terms are those of CHEM, the value of CHEM is
remarkably high in the range of 6.56 to 8.01 ppb/h during the period influenced by the subtropical high and Typhoon Mina
(September 25-October 2, 2019), but only about 2.84 ppb/h during the clean period of October 3-4. In comparison, the
contribution of HTRA in the pollution periods is in the range of -2.74 to -2.81 ppb/h during the day, and about -1.00 to 0.20
ppb/h at night, while the contribution of VTRA in the pollution periods is even smaller in the range of -0.05 to -1.83 ppb/h
during the day, and about -0.12 to -1.19 ppb/h at night. The values in Table 3 clearly show that the enhanced photochemical
production of O₃ is the major cause for the high O₃ concentrations inside PRD and its downwind regions (mostly over the sea
as shown in Figures 7 and 8) during this O₃ episode. Furthermore, this episode accounted for 8 out of a total of 15 days when
hourly O₃ exceeded 93 ppb in PRD during the period of September 1 to October 31, 2019 (Figure 2). Actually only 2 out of
the 15 days occurred in the grey shaded periods (other than subtropical high and typhoon). In addition, this episode
accounted for 10 out of a total of 51 days when MDA8 O₃ exceeded 75 ppb in PRD in the entire year of 2019. Based on the
results above, we propose that the influence of a Pacific subtropical high followed by Typhoon Mina in early autumn 2019 is
the major cause of the most severe O₃ pollution year since the official O₃ observation started in PRD in 2006 (He G et al.,
2021; Li et al., 2022). Moreover, since this O₃ episode is a synoptic scale phenomenon covering the entire eastern China
(The Committee for Ozone Pollution Control, 2020), we also suggest that the enhanced photochemical production of O₃ in
this O₃ episode is a major cause of the high O₃ concentrations observed in eastern China in 2019.

## 4 Summary and conclusions

In late September 2019, a severe O₃ pollution episode with the longest duration since the observation records began occurred
in the PRD. In this study, we have analyzed the effects of individual meteorological and photochemical processes on the O₃
concentration in PRD during this episode by carrying out the WRF-CMAQ model simulations. According to the synoptic
patterns and variations in O₃ concentration, the O₃ episode was divided into three periods: the subtropical high period,
typhoon period, and clean period. By comparing the meteorological parameters at different periods, we found that the
meteorological factors leading to this O₃ pollution episode were low RH, high boundary layer height, predominantly
northerly winds at the surface, and strong downdraft.

From the spatial distribution of O₃ and wind field, it can be seen that the prevailing northerly wind in PRD, induced by the
subtropical high and Typhoon Mina, can transport high concentrations of locally generated O₃ out overseas. In addition,
under the influence of prevailing northerly winds, the occurrence of the sea breeze was delayed until late afternoon, and the
sea breeze that lasted into the night transported the O₃ from the sea back to land. The contribution of horizontal transport to
the increase in O₃ concentration during the influence of sea breeze was about 5.5-8.4 ppb. The end of the episode was due to

the northward movement of Typhoon Mina away from PRD, which resulted in a strong southerly wind, bringing clean and moist oceanic air to PRD. In addition, the temporal-vertical distribution of $O_3$ concentration shows that $O_3$ and its precursors stored in the residual layer above the surface layer at night can be mixed down to the surface by vertical motion in the next morning, thus increasing the daytime ground-level $O_3$ concentration the next day.

The CHEM exhibits the predominant positive contribution to the $O_3$ budget of the boundary layer (0-1260m) in the entire $O_3$ episode, with the remarkably high values in the range of 6.56 to 8.01 ppb/h during the period influenced by the subtropical high and Typhoon Mina, but only about 2.84 ppb/h during the clean period. In comparison, the HTRA and VTRA are the dominant negative contributor to the $O_3$ budget throughout the whole day, with daytime average production rate during the pollution periods of about -2.81ppb/h and -0.94ppb/h, respectively. As this episode accounted for 10 out of the yearly total of

51 days when (MDA8) $O_3$ exceeded 75 ppb in PRD in the entire year of 2019, we propose that the influence of a Pacific subtropical high followed by Typhoon Mina is the major cause of the most severe $O_3$ pollution year since the official $O_3$ observation started in PRD in 2006. Moreover, since this $O_3$ episode occurs not only in the PRD but also in entire eastern China, we also suggest that the increased photochemical production of $O_3$ in this $O_3$ episode is a main reason for the high $O_3$ concentrations observed in eastern China in 2019.

**Data availability**

Hourly surface $O_3$ were obtained from the China National Environmental Centre (http://www.cnemc.cn/en/, last access: 15 April 2022). Hourly meteorological data were provided by the Guangdong Meteorological Service. The ERA5 data were acquired from European Centre for Medium-Range Weather Forecasts Reanalysis v5 dataset (https://cds.climate.copernicus.eu/, last access: 15 April 2022). The FNL meteorological data were taken from the National

Center for Environmental Prediction (https://rda.ucar.edu/, last access: 15 April 2022). Model output data of this paper are available upon request.

**Author Contributions**

 TD and SL proposed the essential research idea. SO, JC and TD performed the model simulations work and carried out the model output data analysis. SO wrote the original paper with input from TD, SL and RL. SL, RL, DT and JL helped revised

the paper. WN and GH discussed the results and offered valuable comments.

**Competing interests**

The authors declare that they have no conflict of interest.



**Acknowledgments**

We would like to acknowledge the China National Environmental Centre, the Guangdong Meteorological Service, the
European Centre for Medium-Range Weather Forecasts and the National Center for Environmental Prediction for providing
datasets that made this work possible. The numerical calculations were performed on the Guangdong Meteorological Service.
We also acknowledge the support of the Institute for Environmental and Climate Research in Jinan University and the
Institute of Tropical and Marine Meteorology/Guangdong Provincial Key Laboratory of Regional Numerical Weather
Prediction, China Meteorological Administration.

**Financial support**

This research was supported by the Guangdong Provincial Key Research and Development Program (grant number
2020B1111360003), the National Natural Science Foundation of China (grant number 41775037, 92044302), Guangzhou
Municipal Science and Technology Project, China (grant number 202002020065), Special Fund Project for Science and
Technology Innovation Strategy of Guangdong Province (grant number 2019B121205004), Guangdong Innovative and
Entrepreneurial Research Team Program (grant number 2016ZT06N263).

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

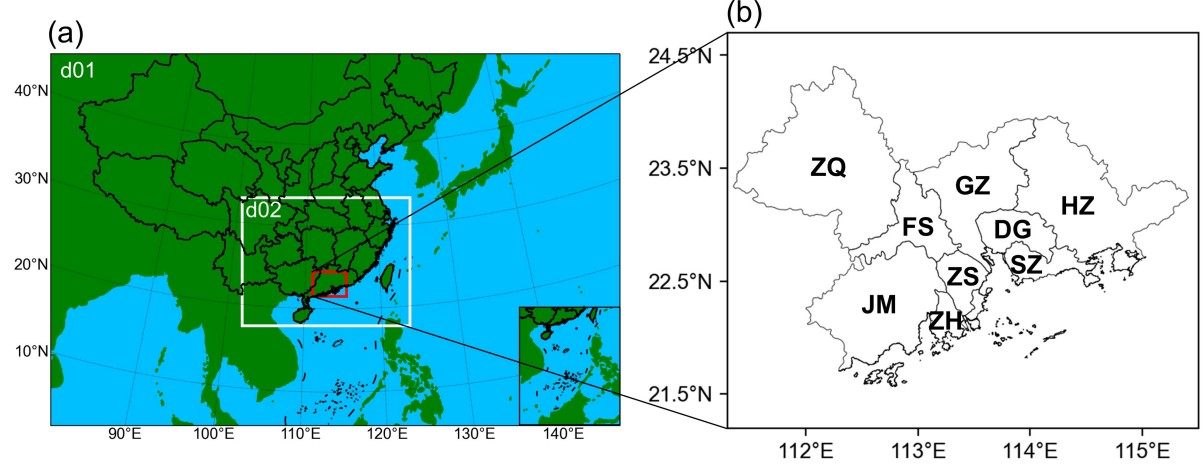


**Figure 1: (a) Two nested model domains in the WRF-CMAQ model and (b) the location of nine main cities in PRD region.**

**Figure 2: (a) Diurnal variation of O₃ concentration in 9 cities (a total of 56 stations averaged) of PRD from September 1 to October 31 2019. The red dotted line indicates the O₃ concentration of 93 ppb. The yellow shade indicates the period affected by the subtropical high; the blue shade indicates the period affected by the typhoon; the grey shade indicates the period affected by the synoptic patterns other than subtropical high and typhoon. (b) Calendar chart of O₃ concentration levels for September and October 2019. These cities are sorted by longitude. Lev1–5 in colored bars represent excellent, good, light pollution, moderate pollution, and severe pollution of air quality index categories in the Technical Regulation Ambient Air Quality Index (AQI), respectively.**


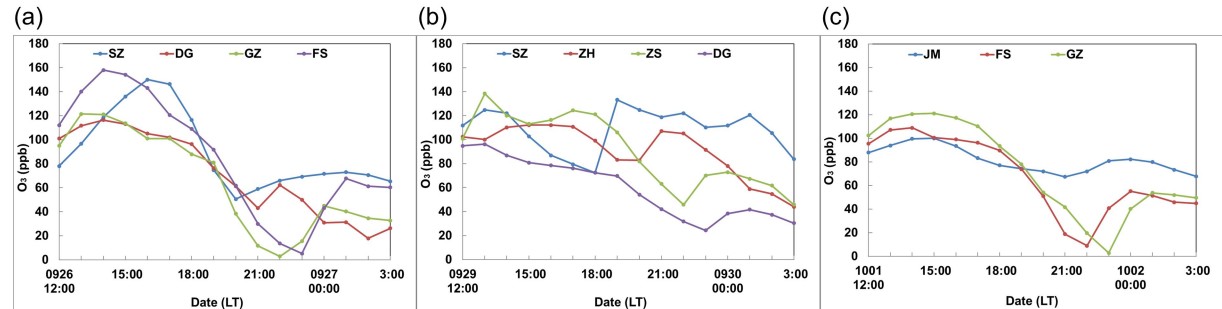

**Figure 3: Time series of sites with nocturnal ground-level O₃ rebound in sequence from coast to inland. (a) Time series in SZ, DG, GZ and FS on September 26. (b) Time series in SZ, ZH, ZS and DG on September 29. (c) Time series in JM, FS and GZ on October 1.**

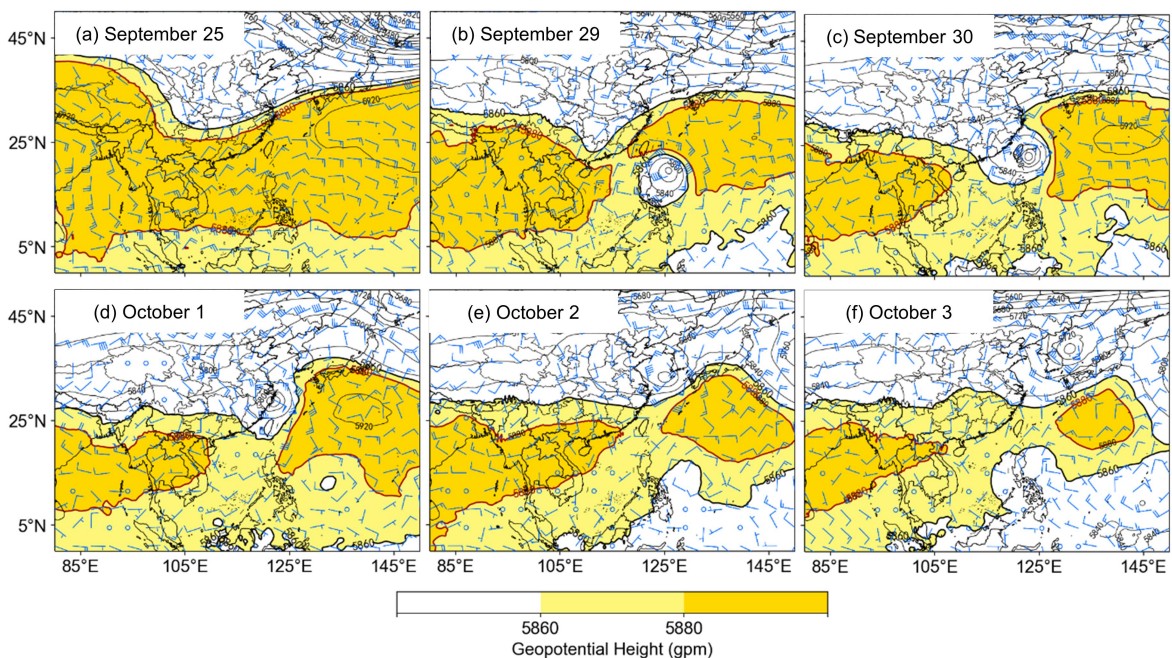

**Figure 4: Spatial distribution of 500 hPa geopotential height and 850hPa wind field over East Asia at 14:00 on September 25 (a) and from September 29 to October 3 (b)-(f).**



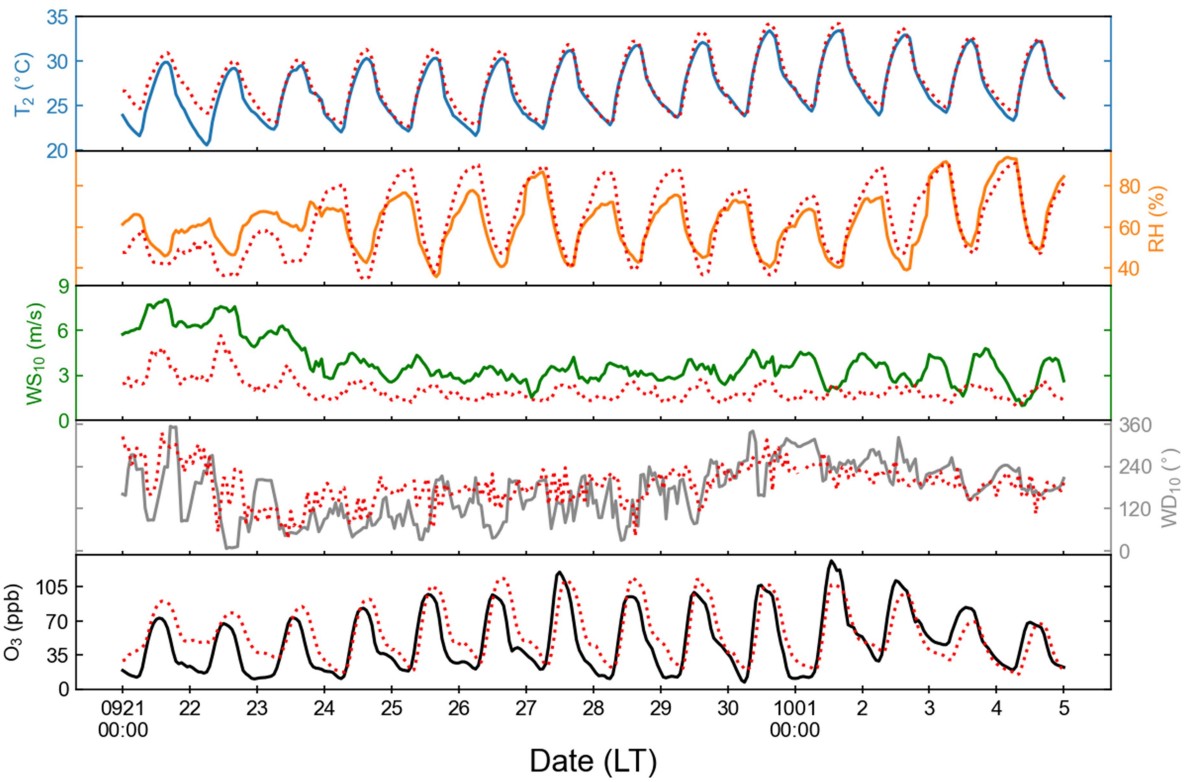

**Figure 5: Hourly variations of T$_2$, RH, WS$_{10}$, WD$_{10}$, and O$_3$ in observed values (red dots) compared to model simulations (solid lines) during September 21 to October 5, 2019.**





**Figure 6: (a) Box plot of different meteorological parameters at the subtropical high period (September 25–28), typhoon period (September 29–October 2), and clean period (October 3–4). (b) Box plot of the comparisons of key meteorological parameters when Typhoon Mina was in different locations. (c) Comparisons of virtual potential temperature ($\theta_v$) profiles in different periods at 14:00. The boxes in (a) and (b) represent interquartile range of each meteorological parameter; the lines dividing the boxes represent the median; the whiskers represent the maximum and minimum values other than outliers; the red dots represent the mean; and the triangles represent the maxima and minima.**





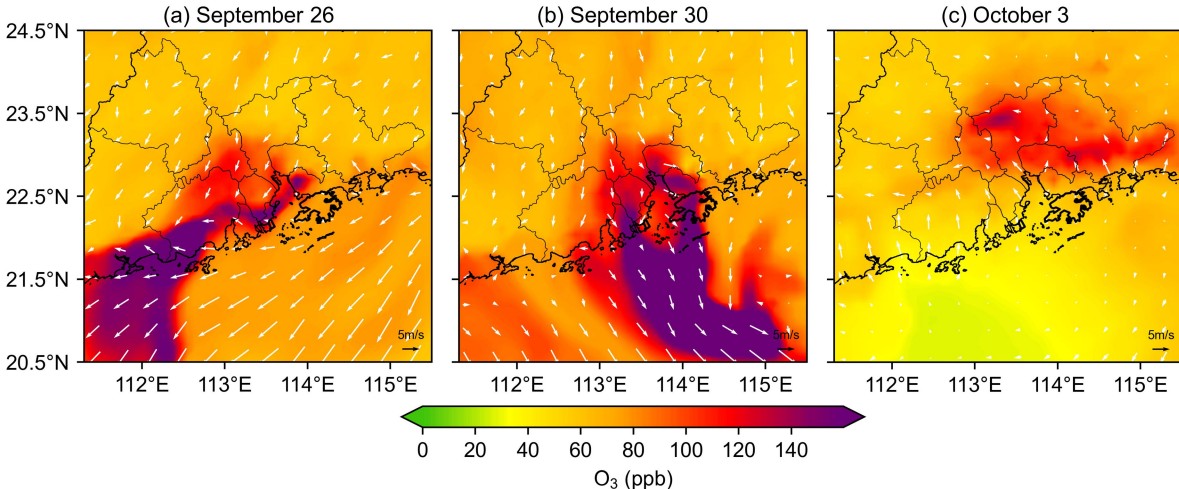

**Figure 7: Spatial distribution of O₃ and wind fields at the ground level at 15:00 on (a) September 26, (b) September 30 and (c) October 3.**

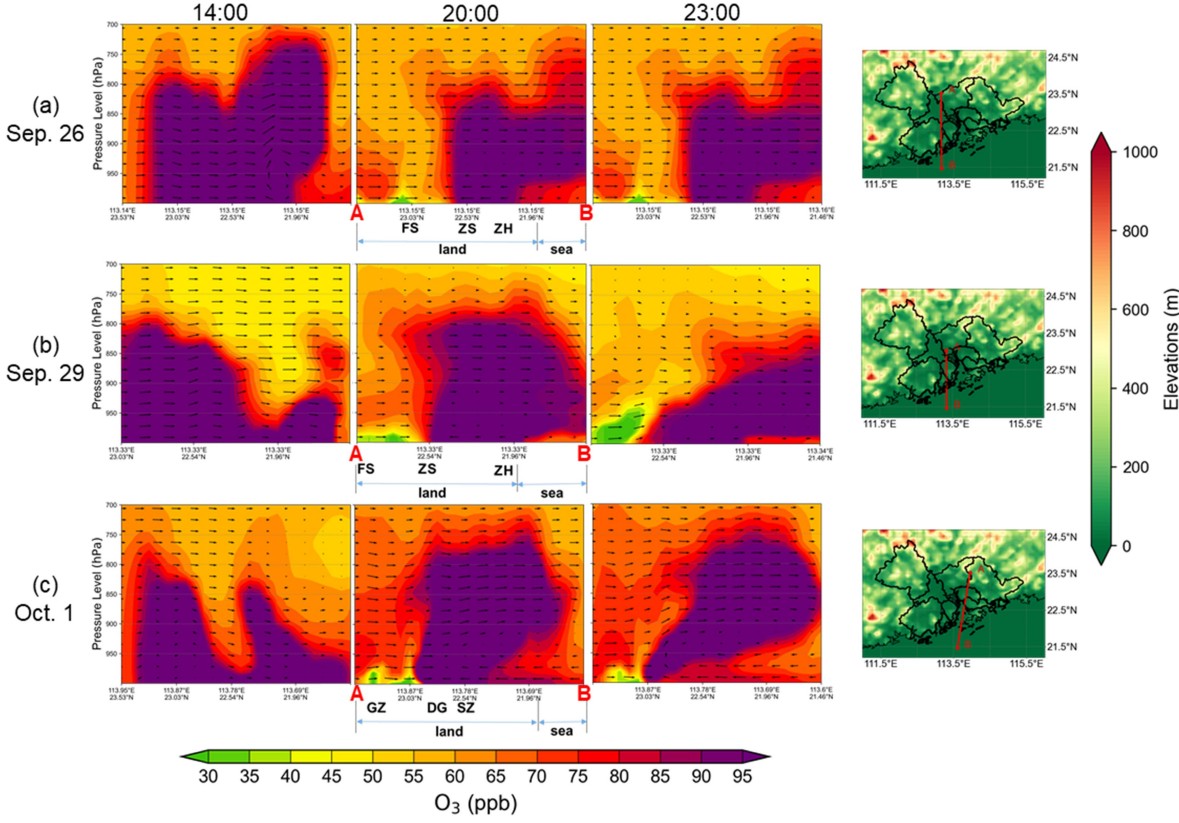

**Figure 8: Vertical distribution of O₃ concentrations (contours) and atmospheric circulation (wind vectors) over PRD along the WD₁₀ at 14:00, 20:00 and 23:00 on (a) September 26, (b) September 29 and (c) October 1.**





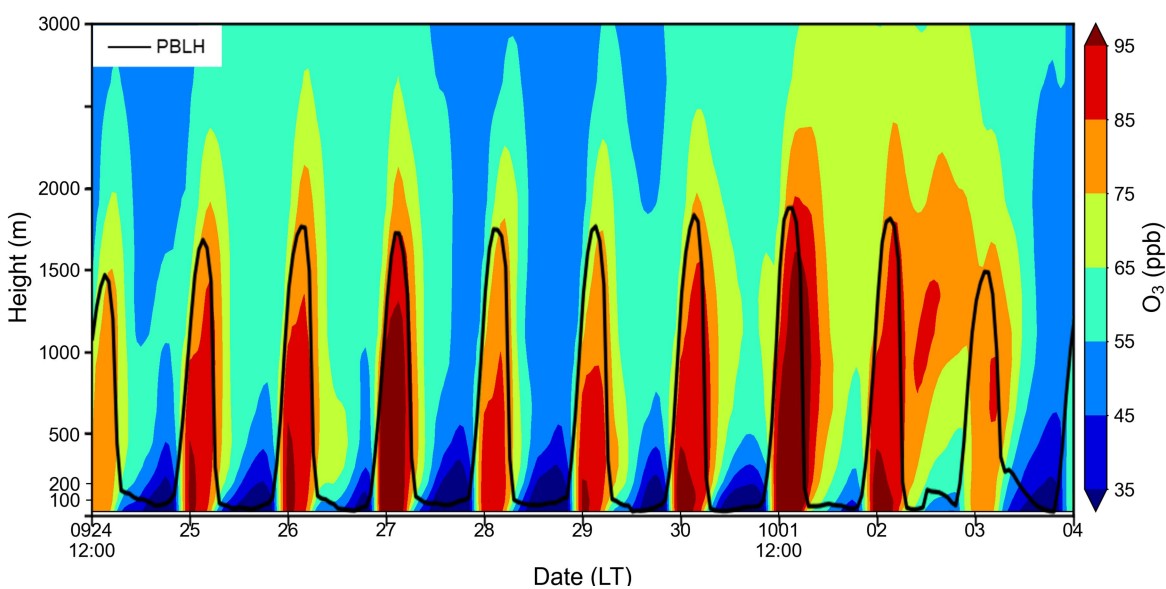

**Figure 9: Temporal-vertical distribution of O₃ concentration above PRD during the O₃ episode. The black line (PBLH) represents the height of the planetary boundary layer.**

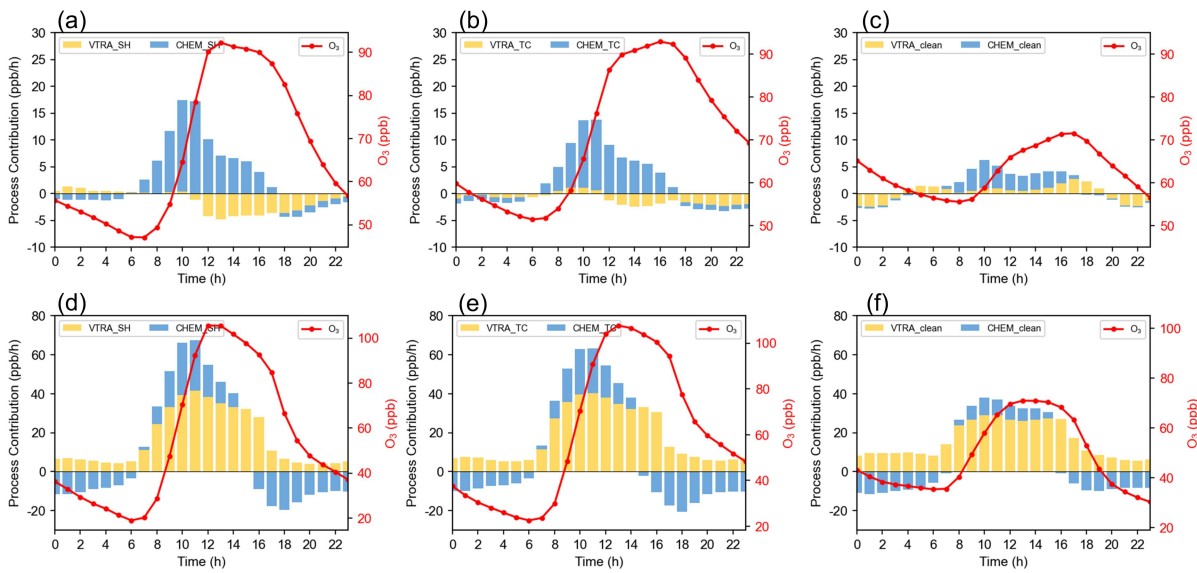

**Figure 10: The O₃ concentrations (red line) and the contributions of photochemical (CHEM, blue bars) and vertical transport (VTRA, yellow bars) processes to O₃ in PRD during the subtropical high period (a, d), typhoon period (b, e) and clean period (c, f), where (a)-(c) are the values for all layers above the near-surface layer (35-1260m), and (d)-(f) are the values for the near-surface layer (0-35m).**





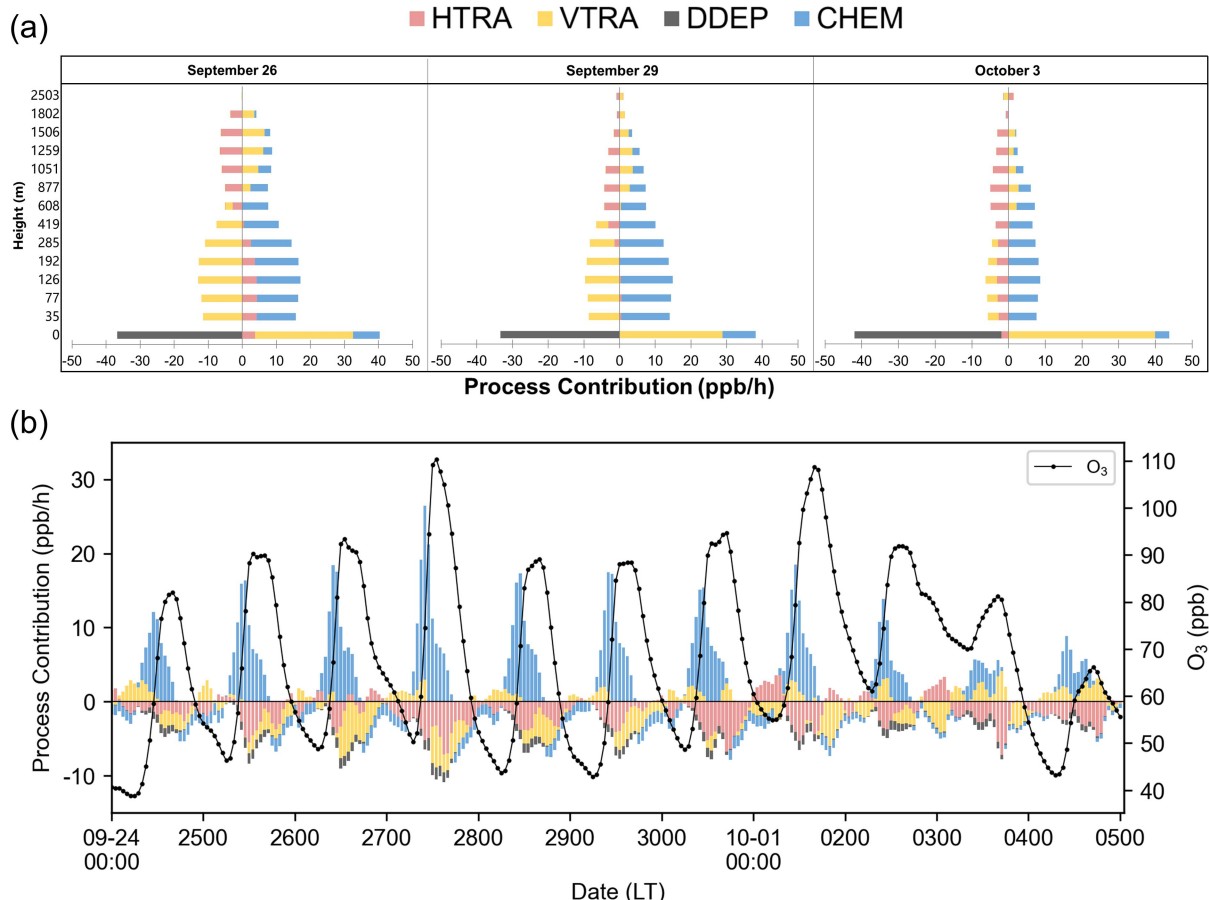

Figure 11: (a) Daytime mean (08:00–18:00) hourly vertical contributions of individual processes to O₃ in PRD under different periods. (b) Time series of individual processes contributing to O₃ in PRD. The black line (O₃) represents the averaged O₃ concentrations under the layers below 1260m.

**Table 1: Physical parameterization configuration options of WRF in this study.**

| Items | Options |
|---|---|
| Microphysics (mp_physics) | WRF Single-Moment 5-class scheme (Hong et al., 2004) |
| Longwave Radiation (ra_lw_physics) | RRTM scheme (Mlawer et al., 1997) |
| Shortwave Radiation (ra_sw_physics) | Goddard shortwave (Kim and Wang, 2011) |
| Surface Layer (sf_sfclay_physics) | Revised MM5 Monin-Obukhov scheme (Jimenez, renamed in v3.6) (Monin and Obukhov, 1954) |





| Land Surface (sf_surface_physics) | Noah Land Surface Model (Chen and Dudhia, 2001) |
| --- | --- |
| Planetary Boundary layer | ACM2 PBL (Pleim. 2007) |
| Cumulus Parameterization (cu_physics) | GD ensemble scheme (Grell and Dévényi, 2002) |

**Table 2: Statistical metrics of meteorological parameters and $O_3$ in the comparison between the observations and simulations during September 21 - October 4.**

|  | obs | sim | *R* | NMB (%) | RMSE | IOA |
| --- | --- | --- | --- | --- | --- | --- |
| $T_2$(°C) | 27.7 | 26.9 | 0.97 | -1.92 | 1.05 | 0.97 |
| RH (%) | 63.6 | 61.8 | 0.84 | -0.97 | 9.10 | 0.90 |
| $WS_{10}$ (m/s) | 2.0 | 3.8 | 0.69 | 69.16 | 2.09 | 0.49 |
| $WD_{10}$ (°) | 188.3 | 170.5 | 0.43 | -9.84 | 74.60 | 0.64 |
| $O_3$ (ppb) | 57.3 | 49.1 | 0.79 | -14.25 | 20.36 | 0.87 |

**Table 3: Daytime (08:00–18:00) and nighttime (19:00–07:00 the next day) mean contributions of individual $O_3$ processes for layers below 1260 m under different periods.**

|  | September 25-28 | | September 29-October 2 | | October 3-4 | |
| --- | --- | --- | --- | --- | --- | --- |
|  | daytime | nighttime | daytime | nighttime | daytime | nighttime |
| $O_3$ | 81.53 | 56.56 | 83.04 | 68.30 | 67.03 | 55.01 |
| CHEM | 8.01 | -1.02 | 6.56 | -0.85 | 2.84 | -0.44 |
| VTRA | -1.83 | -0.12 | -0.05 | -1.19 | 1.80 | -0.30 |
| HTRA | -2.74 | -1.00 | -2.81 | 0.20 | -2.88 | -0.86 |
| DDEP | -0.96 | -0.05 | -0.87 | -0.08 | -0.74 | -0.06 |