# Peer review of "Impact of a subtropical high and a typhoon on a severe ozone pollution episode in the Pearl River Delta, China"

_Atmospheric Chemistry and Physics, 2022_

## Referee Comment (RC2)

"Impact of a subtropical high and a typhoon on a severe ozone pollution episode in the Pearl River Delta, China" by Shanshan Ouyang et al. discussed in detail the influence of a subtropical high and a typhoon weather process on severe O3 pollution in the Pearl River Delta. The manuscript provides valuable information on the formation mechanism of ozone pollution in coastal areas under such weather conditions. There are some minor suggestions before publication.

General comments

In Section 3.2, why the correlation between T and RH simulation results is as high as 0.97 and 0.84, while the correlation between WD10 and WS10 simulation results is only 0.69 and 0.64, can you give some explanation?
In Section 3.3, the meteorological factors that are favorable to the development of ozone-polluted weather during the typhoon period and the subtropical high pressure period are the same? so why? Can you compare the two periods separately?

Specific comments

In the introduction, please pay attention to the tense.

Line 72: Summary and conclusions are presented in Section 4

Figure 11(b): Please modify the abscissa of Figure 11(b). For example, change 2500 to 25.

---

## Author Comment (AC1)

Dear Editor,

We appreciate the prompt reviews and would like to thank the reviewers for insightful comments and suggestions on our manuscript entitled "Impact of a subtropical high and a typhoon on a severe ozone pollution episode in the Pearl River Delta, China" (MS No.: acp-2022-290). We have carefully considered all comments and suggestions. Listed below are our point-by-point responses to all comments and suggestions of this reviewer (Reviewer's points in black, our responses in blue).

**Anonymous Referee #1**

The manuscript entitled "Impact of a subtropical high and a typhoon on a severe ozone pollution episode in the Pearl River Delta, China" by Shanshan Ouyang et al. explored in details how the severe $O_3$ pollution in PRD is influenced by the weather system of subtropical high and typhoon. The manuscript provides valuable information for understanding the ozone pollution formation mechanism in coastal areas, and is well within the scope of ACP. I only have the following minor comments needed to be addressed before the publication.

**Response:**

We appreciate the encouraging comments and suggestions.

**General comments**

One of the major findings of this manuscript is that the photochemical $O_3$ production is enhanced during the influence of subtropical high and typhoon, and acts a major cause of the most severe $O_3$ pollution in PRD. However, why the photochemistry process is enhanced during the two events is not clearly discussed. Especially, How the enhanced photochemistry related to changed meteorological factors? Although the meteorological factors and photochemical process are separately discussed, there are inner relationship between meteorological factors and photochemical process. I suggest to further elucidate how the changes in meteorological factors induced by

typhoon and the subtropical high influences the photochemical production of ozone.

**Response:**

Thank you for the insightful comments and helpful suggestions. It is an oversight that we only emphasized the main meteorological factors conducive to $O_3$ pollution under the influence of subtropical high and typhoon, but ignored the inter relationship between the changes of meteorological factors and photochemical enhancement. We now have further elucidated how the changes in meteorological factors induced by the subtropical high and the typhoon influence the photochemical production of $O_3$ in Section 3.3 (page 7 to 8, line 200 to 230) and Section 3.5 (page 11, line 330 to 333) of the revised manuscript, which are reproduced below for your information.

**Page 7 to 8, line 200 to 230:** SWDOWN was strong throughout the $O_3$ episode, and the resulting high $NO_2$ photolysis rate provided sufficient conditions for the generation of $O_3$, especially in the upper and middle layers of the boundary layer (Dickerson, 1997; He et al., 2021). Compared to the clean period, lower RH, higher PBLH, predominantly weak northerly winds at the surface (negative $V_{10}$) and stronger downdraft (positive Omega) were found in the first two periods. Lower RH tends to be unfavorable for the wet deposition of $O_3$ (He et al., 2017; Han et al., 2019; Li et al., 2021). The relatively high thermal PBLH allows for adequate mixing of $O_3$ (Li et al., 2018; Zhao et al.,2019; Dong et al., 2020), which is more conducive to the downward transport of $O_3$ in the upper layer when superimposed with the stronger downdraft under the background of the subtropical high and typhoon (Li et al., 2022; Liu et al., 2022). The weak northerly winds favor the transport of high concentrations of locally generated $O_3$ southwards to the coastal areas, resulting in higher $O_3$ concentrations in the south of PRD than in the north during the pollution periods. At the same time, the stable weather under the background of the subtropical high and Typhoon Mina are more favorable for the formation of a deep residual layer, which can store photochemically generated $O_3$ in the daytime and exacerbate the surface $O_3$ concentration through vertical transport in the next day. Furthermore, since the virtual

potential temperature ($\theta_v$) can represent the height of the atmospheric mixed layer, it can also be seen from the $\theta_v$ profile results of PRD from September 25 to October 4 at 14:00 in Figure 6c that the inflection point of the $\theta_v$ in the pollution periods are above 1500 m, that is, the mixed layers are higher, indicating that the relatively higher mixed layer height is more conducive to the mixing of $O_3$.

On the other hand, when PRD was under the influence of Typhoon Mina, it had a higher $T_2$, lower RH, a switch to weak northwesterly winds and stronger Omega compared to the subtropical high period, indicating that the more severe meteorological conditions combined with $O_3$ and its precursors accumulated in the subtropical high period were beneficial to the further enhancement of $O_3$ photochemical generation. Further, the key meteorological parameters affecting the changes in $O_3$ concentration in PRD varied as Typhoon Mina moved away from PRD can be seen in Figure 6b: $T_2$, $W_{10}$, weak northerly wind (negative $V_{10}$) and Omega all rose first and then gradually decreased after October 1, indicating that the meteorological conditions were more favorable for $O_3$ generation and accumulation when PRD was the influence of typhoon peripheral circulation.

The findings above suggest that the meteorological factors such as lower RH, predominantly weak northerly winds, stronger downdrafts and higher PBLH caused by the subtropical high and Typhoon Mina were the main reasons for the development of this $O_3$ pollution episode, and it ended due to the switch to clean southerly winds at lower levels.

**page 11, line 330 to 333:** Again, the dominant terms are those of CHEM, the value of CHEM is remarkably high in the range of 6.56 to 8.01 ppb/h during the period influenced by the subtropical high and Typhoon Mina (September 25 to October 2, 2019) due to strong photochemical reaction rates resulting from higher SWDOWN and lower RH, but only about 2.84 ppb/h during the clean period of October 3-4 due to the switch to clean southerly winds.

**Specific comments**

1. Line 114: Is the $O_3$ concentration corresponding to the simulated $O_3$ in the lowest layer (i.e., below 35 m)? If take the lowest 3 layers into account, especially for periods strongly influenced by downdraft, what would the comparison between the model simulation and the observation look like?

**Response:**

Yes, the simulated $O_3$ in Figure 5 was obtained from the model's lowest layer, and we have explained it in the revised manuscript (page 6, line 183). As shown in Figure R1 and Table R1, when the lowest 3 layers are combined into one, the comparison between the model simulation and the observation would be closer and the evaluation results would be better. Combined with the daytime vertical contributions of individual processes to $O_3$ in Figure 11a, it could be seen that since the photochemical generation of $O_3$ was more intense above the near-surface layer, considering the results of the lowest 3 layers would reduce the negative deviation between the simulation and observation.

[Figure]

Figure R1. Hourly variations of $O_3$ in observed values (blue dots) compared to model simulations of the lowest layer (solid black line) and lowest 3 layers (solid red line) from September 21 to October 4, 2019.

Table R1. Statistical metrics of $O_3$ in the comparison between the observations and simulations (from lowest layer and lowest 3 layers) from September 21 to October 4,

2019.

| | obs | sim | $R$ | NMB (%) | RMSE | IOA |
|---|---|---|---|---|---|---|
| $O_3$ (ppb, from lowest layer) | 57.3 | 49.1 | 0.79 | -14.25 | 20.36 | 0.87 |
| $O_3$ (ppb, from lowest 3 layers) | 57.3 | 52.2 | 0.86 | -8.66 | 15.85 | 0.92 |

2. Line 115: Please define "NAWO" and "CNMC". Is CNMC the same as NEMC in Line 76? If so, please keep the abbreviation consistent.

**Response:**

Thanks for pointing out our negligence. We have described the sources of $O_3$ data (line 77) and meteorological data (line 84) in section 2.1 of the revised manuscript with new abbreviations. At the same time, the sentence in line 117 is modified by removing the two abbreviations ("NAWO" and "CNMC") for simplicity.

3. Line 133: Please add description on "the second standard of air quality".

**Response:**

Thanks for pointing out our negligence. For hourly $O_3$ concentration, the national ambient air quality secondary standard is 200 μg m$^{-3}$ (approximately 93 ppb). We have described it in line 42 of the revised manuscript.

4. Line 137 - 138: Please define "Lev 3" and "Lev 5".

**Response:**

Thanks again for pointing out our negligence. We have added the descriptions of "Lev 1-5" in line 137-139 of the revised manuscript.

5. Line 173: The model overestimated WS$_{10}$ quite a lot. Which of the wind vector (i.e.,

u, v, ω) has not been well reproduced that could lead to the $WS_{10}$ overestimation? How would this overestimation further influence the evaluation of the contribution of transport / sea breeze?

**Response:**

The model does overestimate $WS_{10}$ quite a lot and we acknowledge this referee's concerns that the overestimated model $WS_{10}$ might further influence the subsequent analysis. For mesoscale models such as WRF, the simulation of the near-surface wind fields under stable weather background is still a challenge (Lim et al., 2018; Srivastava et al., 2021), with most of the previous simulations of near-surface wind speeds being on the high side (Tymvios et al., 2018; Lorenz et al., 2022). We evaluated the variables $U_{10}$ and $V_{10}$ which were directly related to $WS_{10}$ to explore whether the overestimation of model $WS_{10}$ would affect the subsequent analysis of pollution processes. As can be seen from Figure R2 and Table R2, although the simulation of $U_{10}$ also has a certain negative deviation with a mean bias (MB) of -0.26m/s, the main reason for the overestimation of $WS_{10}$ is the larger negative deviation of the $V_{10}$ simulation (MB of -1.41m/s). That is, the northerly wind simulation is too large, which may overestimate the contribution of the prevailing winds in transporting locally generated $O_3$ in PRD to the sea during the influence of the subtropical high and Typhoon Mina. Although the positive $V_{10}$ simulations on October 3-5 are large and may overestimate the contribution of clean sea breeze to this $O_3$ episode, the model well simulates the nighttime sea breeze during the pollution periods, providing certain degree of reliability for subsequent studies of the effect of sea breeze on $O_3$. Future studies are planned to improve the simulation of near-surface wind fields by using more detailed terrain data and more suitable parameterization schemes.

[Figure]

Figure R2. Hourly variations of $U_{10}$ and $V_{10}$ in observed values (red dots) compared to model simulations (solid lines) during September 21 to October 4, 2019.

Table R2. Statistical metrics of $U_{10}$ and $V_{10}$ in the comparison between the observations and simulations during September 21 to October 4, 2019.

|  | obs | sim | $R$ | MB | RMSE | IOA |
|---|---|---|---|---|---|---|
| $U_{10}$ (m/s) | 0.1 | -0.2 | 0.67 | -0.26 | 1.30 | 0.66 |
| $V_{10}$ (m/s) | -0.5 | -2.0 | 0.84 | -1.41 | 2.23 | 0.73 |

6. Line 189: Please define $U_{10}$, $V_{10}$.

**Response:**

Thanks. We have defined Zonal wind speed at 10m ($U_{10}$) and Meridional wind speed at 10m ($V_{10}$) in line 193 of the revised manuscript.

7. Line 210: Please define $\theta_v$.

**Response:**

Thanks. We have added the definition of virtual potential temperature ($\theta_v$) in line 212 of the revised manuscript.

8. Line 213 - 216: Higher PBLH could result in higher $O_3$ concentration due to enhanced contribution of downward $O_3$ transport. However, higher PBLH could also favor the dilution of $O_3$ and its precursors, thus result in weaker $O_3$ production and accumulation. What would be the balance between these two effects?

**Response:** The referee made a very insightful point that higher PBLH may facilitate downward transport of upper $O_3$, but may also favor the dilution of $O_3$ and its precursors. However, for the thermal boundary layer affected by surface thermodynamics, the stronger the solar radiation or the higher the air temperature, the higher the PBLH, and the meteorological conditions at that time tend to favor the photochemistry of ozone. Previous studies have shown that high PBLH was often accompanied by high concentrations of $O_3$ (Zhao et al., 2019; Dong et al, 2020; He et al., 2021; Liu et al., 2022). In this study, as can be seen from the PBLH heights for each period in Figure 6a and the results from Table 3, the PBLH was higher during the two pollution periods (September 25-28, September 29 to October 2), and the $O_3$ concentration within the boundary layer was also higher during these periods, with 81.53 ppb and 83.04 ppb, respectively. In these cases, the VTRA were -1.83 ppb/h and -0.05 ppb/h, respectively, indicating that higher $O_3$ concentrations within the PBL resulted in vertically upward transport of $O_3$ to the upper atmosphere. In contrast, the daytime PBLH on October 3-4 was lower and a VTRA of 1.80 ppb/h, i.e. $O_3$ above the PBL was transported downward. Meanwhile, photochemical production of $O_3$ was stronger when the boundary layer was higher during the two pollution periods, with CHEM of 8.01 ppb/h and 6.56 ppb/h, respectively, but was only 2.84 ppb/h in during the clean period (October 3-4), suggesting a negligible dilution effect. In summary, under the influence of a subtropical high and Typhoon Mina, high PBLH and associated meteorological conditions were highly conducive to the photochemical

production of $O_3$ in the boundary layer in PRD, which resulted in extremely high $O_3$ concentrations within the PBL and a net vertical $O_3$ transport to the upper atmosphere.

9. Figure 5: It looks like there is a 1-hour time shift between the simulated and the observed $O_3$ concentration. This could be caused by the definition of the measurement time of the CNEC $O_3$ data (i.e., data at 1:00 represent the averages in 0:00 – 1:00). If this time shift has been taken into account, would the discrepancy between the simulation and the observation become smaller?

**Response:**

We are grateful for the referee's thoughtful point on "the 1-hour time shift between the simulated and the observed $O_3$ concentration could be caused by the definition of the measurement time of the observed $O_3$ data". We looked through the *Specifications and Test Procedures for Ambient Air Quality Continuous Automated Monitoring System for SO2、NO2、O3 and CO* (HJ 654-2013) developed by China National Environmental Monitoring Center (CNEMC). The $O_3$ data reported at a certain hour was indeed the average of measurements for the hour before (e.g. data at 1:00 represent the average values in 0:01 – 1:00). In comparison, the *Operational Guidance for the Community Multiscale Air Quality (CMAQ) Modeling System CMAQ Version 4.7.1* shows that "the 2-D CCTM integral average concentration file (ACONC) contains average model species concentrations for each model hour, as opposed to instantaneous concentrations at the end of each output time step", and line 174 in the source code *wr_aconc.F,v* for calculating ACONC under the path of */cmaq4.7.1/models/CCTM/src/driver/ctm/* shows that "Timestamp represents beginning computed date/time". That is, the ACONC $O_3$ concentration at a certain hour used in the simulation evaluation in this paper was the average of the model values for the hour after the reported time. Therefore, there is indeed a 1-hour time shift between the simulated and the observed $O_3$ concentration. As can be seen from Figure R3 and Table R3, if this time shift is taken into account, the discrepancy between the simulation and the observation does become smaller, with *R* reaching

0.89. We have corrected the corresponding parts of the figures (Figure 5 and Figure 7) and evaluation (Table 2) that using the ACONC data in the revised manuscript.

[Figure]

Figure R3. Hourly variations of $O_3$ in observed values (blue dots) compared to model simulations without time shift (solid back line) and with a 1-hour time shift forward (solid red line) from September 21 to October 4, 2019.

Table R3. Statistical metrics of $O_3$ in the comparison between the observations and simulations (without time shift and with a 1-hour time shift) from September 21 to October 4, 2019.

|  | obs | sim | $R$ | NMB (%) | RMSE | IOA |
|---|---|---|---|---|---|---|
| $O_3$ (ppb, without time shift) | 57.3 | 49.1 | 0.79 | -14.25 | 20.36 | 0.87 |
| $O_3$ (ppb, with a 1-hour time shift) | 57.3 | 49.0 | 0.89 | -14.25 | 16.15 | 0.92 |

10. Figure 11b: Do the individual processes correspond to the averages of whole boundary layer?

**Response:**

Thanks for pointing out our negligence. As can be seen in section 3.5 in line 308, the individual processes do correspond to the averages of the whole boundary layer. We have changed accordingly in the revised manuscript (page 26, line 617).

**References**

Dickerson, R. R.: The impact of aerosols on solar ultraviolet radiation and photochemical smog, Science, 278, 827–830, https://doi.org/10.1126/science.278.5339.827, 1997.

Dong, Y., Li, J., Guo, J., Jiang, Z., Chu, Y., Chang, L., Yang, Y., and Liao, H.: The impact of synoptic patterns on summertime ozone pollution in the North China Plain, Sci. Total Environ., 735, 139559, https://doi.org/10.1016/j.scitotenv.2020.139559, 2020.

Han, Y., Gong, Z., Ye, J., Liu, P., McKinney, K. A., and Martin, S. T.: Quantifying the role of the relative humidity-dependent physical state of organic particulate matter in the uptake of semivolatile organic molecules, Environ. Sci. Technol., 53, 13209–13218, https://doi.org/10.1021/acs.est.9b05354, 2019.

He, G., Deng, T., Wu, D., Wu, C., Huang, X., Li, Z., Yin, C., Zou, Y., Song, L., Ouyang, S., Tao, L., and Zhang, X.: Characteristics of boundary layer ozone and its effect on surface ozone concentration in Shenzhen, China: A case study, Sci. Total Environ., 791, 148044, https://doi.org/10.1016/j.scitotenv.2021.148044, 2021.

He, X., Pang, S., Ma, J., and Zhang, Y.: Influence of relative humidity on heterogeneous reactions of $O_3$ and $O_3/SO_2$ with soot particles: Potential for environmental and health effects, Atmos. Environ., 165, 198–206, https://doi.org/10.1016/j.atmosenv.2017.06.049, 2017.

Li, K., Jacob, D. J., Liao, H., Shen, L., Zhang, Q., and Bates, K. H.: Anthropogenic drivers of 2013–2017 trends in summer surface ozone in China, P. Natl. Acad. Sci. USA, 116, 422–427, https://doi.org/10.1073/pnas.1812168116, 2018.

Li, M., Yu, S., Chen, X., Li, Z., Zhang, Y., Wang, L., Liu, W., Li, P., Lichtfouse, E., Rosenfeld, D., and Seinfeld, J. H.: Large scale control of surface ozone by relative

humidity observed during warm seasons in China, Environ. Chem. Lett., 19, 3981–3989, https://doi.org/10.1007/s10311-021-01265-0, 2021.

Li, Y., Zhao, X., Deng, X., and Gao, J.: The impact of peripheral circulation characteristics of typhoon on sustained ozone episodes over the Pearl River Delta region, China, Atmos. Chem. Phys., 22, 3861–3873, https://doi.org/10.5194/acp-22-3861-2022, 2022.

Lim, K.-S. S., Lim, J.-M., Shin, H. H., Hong, J., Ji, Y.-Y., and Lee, W.: Impacts of subgrid-scale orography parameterization on simulated atmospheric fields over Korea using a high-resolution atmospheric forecast model, Meteorol. Atmos. Phys., 131, 975–985, https://doi.org/10.1007/s00703-018-0615-4, 2018.

Liu, H., Han, X., Tang, G., Zhang, J., Xia, X., Zhang, M., and Meng, L.: Model analysis of vertical exchange of boundary layer ozone and its impact on surface air quality over the North China Plain, Sci. Total Environ., 821, 153436, https://doi.org/10.1016/j.scitotenv.2022.153436, 2022.

Lorenz, T., Mayer, S., Kral, S. T., Suomi, I., Steeneveld, G. -J., and Holtslag, A. A. M.: The stable atmospheric boundary layer over snow-covered sea ice: Model evaluation with fine-scale ISOBAR18 observations, Q. J. Roy. Meteor. Soc., 148, 2031–2046, https://doi.org/10.1002/qj.4293, 2022.

Madala, S., Salinas, S. V., Wang, J., and Liew, S. C.: Customization of the Advanced Research Weather Research and Forecasting model over the Singapore region: impact of planetary boundary layer schemes, land use, land cover and model horizontal grid resolution, Meteorol. Appl., 26, 221–231, https://doi.org/10.1002/met.1755, 2019.

Srivastava, P., Sharan, M., and Kumar, M.: A note on surface layer parameterizations in the weather research and forecast model, Dynam. Atmos. Oceans, 96, 101259, https://doi.org/10.1016/j.dynatmoce.2021.101259, 2021.

Tymvios, F., Charalambous, D., Michaelides, S., and Lelieveld, J.: Intercomparison of boundary layer parameterizations for summer conditions in the eastern Mediterranean island of Cyprus using the WRF - ARW model, Atmos. Res., 208, 45–59, https://doi.org/10.1016/j.atmosres.2017.09.011, 2018.

Wu, C., Luo, K., Wang, Q., and Fan, J.: Simulated potential wind power sensitivity to the planetary boundary layer parameterizations combined with various topography datasets in the weather research and forecasting model, Energy, 239, 122047, https://doi.org/10.1016/j.energy.2021.122047, 2022.

Zhao, W., Tang, G., Yu, H., Yang, Y., Wang, Y., Wang, L., An, J., Gao, W., Hu, B., Cheng, M., An, X., Li, X., and Wang, Y.: Evolution of boundary layer ozone in Shijiazhuang, a suburban site on the North China Plain, J. Environ. Sci., 83, 152–160, https://doi.org/10.1016/j.jes.2019.02.016, 2019.

---

## Author Comment (AC2)

Dear Editor,

We appreciate the prompt reviews and would like to thank the reviewers for insightful comments and suggestions on our manuscript entitled "Impact of a subtropical high and a typhoon on a severe ozone pollution episode in the Pearl River Delta, China" (MS No.: acp-2022-290). We have carefully considered all comments and suggestions. Listed below are our point-by-point responses to all comments and suggestions of this reviewer (Reviewer's points in black, our responses in blue).

**Anonymous Referee #2**

"Impact of a subtropical high and a typhoon on a severe ozone pollution episode in the Pearl River Delta, China" by Shanshan Ouyang et al. discussed in detail the influence of a subtropical high and a typhoon weather process on severe $O_3$ pollution in the Pearl River Delta. The manuscript provides valuable information on the formation mechanism of ozone pollution in coastal areas under such weather conditions. There are some minor suggestions before publication.

**Response:**

We appreciate the encouraging comments and suggestions.

**General comments**

1. In Section 3.2, why the correlation between T and RH simulation results is as high as 0.97 and 0.84, while the correlation between $WD_{10}$ and $WS_{10}$ simulation results is only 0.69 and 0.64, can you give some explanation?

**Response:**

We appreciate this important comment. Since $T_2$ and RH themselves have regular diurnal variations, the WRF model tends to simulate them well. While the simulation of wind fields is affected strongly by terrains which tend to have large uncertainty. For mesoscale models such as WRF, the simulation of the near-surface wind fields under

stable weather background is still a challenge due to uncertainties in the topography datasets (Wang et al., 2015; Lim et al., 2018; Wu et al., 2022) and differences in boundary layer parameterization schemes (Tymvios et al., 2018; Madala et al., 2019; Srivastava et al., 2021). Future studies are planned to improve the simulation of near-surface wind fields by using more detailed terrain data and more suitable boundary layer parameterization schemes.

2. In Section 3.3, the meteorological factors that are favorable to the development of ozone-polluted weather during the typhoon period and the subtropical high pressure period are the same? so why? Can you compare the two periods separately?

**Response:**

Thank you for the thoughtful comment. In general, both the typhoon periphery and the subtropical high bring sunny and dry weather conducive to $O_3$ production, but the differences in the position of the subtropical high and the typhoon will cause different changes in meteorological factors. As can be seen from the comparison of the different periods in Figure 6a, the pollution periods have lower RH, higher PBLH and stronger downdraft compared to the clean period, which is more conducive to photochemical production of $O_3$. Although both are considered as pollution periods, the meteorological factors under the influence of Typhoon Mina are significantly different from those under the influence of the subtropical high. As shown in Figure 6b, when PRD was under the influence of Typhoon Mina, it had a higher $T_2$, a switch to weak northwesterly winds and stronger Omega compared to the subtropical high period, indicating that the former has more severe meteorological conditions for $O_3$ photochemical generation than the latter. More detailed comparison has been added in section 3.3 of the revised manuscript.

**Specific comments**

1. In the introduction, please pay attention to the tense.

**Response:**

Thank you for the careful reading and for pointing out our tense errors. We have reviewed the introduction and corrected some of the tense mistakes in the revised manuscript.

2. Line 72: Summary and conclusions are presented in Section 4

**Response:**

Thank you. We have corrected it in the revised manuscript.

3. Figure 11(b): Please modify the abscissa of Figure 11(b). For example, change 2500 to 25.

**Response:**

Thank you. We have modified the abscissa accordingly in Figure 11(b) in the revised manuscript.

**References**

Lim, K.-S. S., Lim, J.-M., Shin, H. H., Hong, J., Ji, Y.-Y., and Lee, W.: Impacts of subgrid-scale orography parameterization on simulated atmospheric fields over Korea using a high-resolution atmospheric forecast model, Meteorol. Atmos. Phys., 131, 975–985, https://doi.org/10.1007/s00703-018-0615-4, 2018.

Madala, S., Salinas, S. V., Wang, J., and Liew, S. C.: Customization of the Advanced Research Weather Research and Forecasting model over the Singapore region: impact of planetary boundary layer schemes, land use, land cover and model horizontal grid resolution, Meteorol. Appl., 26, 221–231, https://doi.org/10.1002/met.1755, 2019.

Srivastava, P., Sharan, M., and Kumar, M.: A note on surface layer parameterizations in the weather research and forecast model, Dynam. Atmos. Oceans, 96, 101259, https://doi.org/10.1016/j.dynatmoce.2021.101259, 2021.

Tymvios, F., Charalambous, D., Michaelides, S., and Lelieveld, J.: Intercomparison of boundary layer parameterizations for summer conditions in the eastern Mediterranean island of Cyprus using the WRF - ARW model, Atmos. Res., 208, 45–59, https://doi.org/10.1016/j.atmosres.2017.09.011, 2018.

Wang, N., Guo, H., Jiang, F., Ling, Z. H., and Wang, T.: Simulation of ozone formation at different elevations in mountainous area of Hong Kong using WRF-CMAQ model, Sci. Total Environ., 505, 939–951, https://doi.org/10.1016/j.scitotenv.2014.10.070, 2015.

Wu, C., Luo, K., Wang, Q., and Fan, J.: Simulated potential wind power sensitivity to the planetary boundary layer parameterizations combined with various topography datasets in the weather research and forecasting model, Energy, 239, 122047, https://doi.org/10.1016/j.energy.2021.122047, 2022.

---

## Author Response (AR1)

**Dear Editor,**

We appreciate the prompt reviews and would like to thank the reviewers for insightful comments and suggestions on our manuscript entitled "Impact of a subtropical high and a typhoon on a severe ozone pollution episode in the Pearl River Delta, China" (MS No.: acp-2022-290). We have carefully considered all comments and suggestions. Listed below are our point-by-point responses to all comments and suggestions of this reviewer (Reviewer's points in black, our responses in blue).

**Anonymous Referee #1**

The manuscript entitled "Impact of a subtropical high and a typhoon on a severe ozone pollution episode in the Pearl River Delta, China" by Shanshan Ouyang et al. explored in details how the severe O3 pollution in PRD is influenced by the weather system of subtropical high and typhoon. The manuscript provides valuable information for understanding the ozone pollution formation mechanism in coastal areas, and is well within the scope of ACP. I only have the following minor comments needed to be addressed before the publication.

**Response:**

We appreciate the encouraging comments and suggestions.

**General comments**

One of the major findings of this manuscript is that the photochemical O3 production is enhanced during the influence of subtropical high and typhoon, and acts a major cause of the most severe O3 pollution in PRD. However, why the photochemistry process is enhanced during the two events is not clearly discussed. Especially, How the enhanced photochemistry related to changed meteorological factors? Although the meteorological factors and photochemical process are separately discussed, there are inner relationship between meteorological factors and photochemical process. I suggest to further elucidate how the changes in meteorological factors induced by typhoon and the subtropical high influences the photochemical production of ozone.

**Response:**

Thank you for the insightful comments and helpful suggestions. It is an oversight that we only emphasized the main meteorological factors conducive to  $O_3$  pollution under the influence of subtropical high and typhoon, but ignored the inter relationship between the changes of meteorological factors and photochemical enhancement. We now have further elucidated how the changes in meteorological factors induced by the subtropical high and the typhoon influence the photochemical production of  $O_3$  in Section 3.3 (page 7 to 8, line 200 to 230) and Section 3.5 (page 11, line 330 to 333) of the revised manuscript, which are reproduced below for your information.

Page 7 to 8, line 200 to 230: SWDOWN was strong throughout the O3 episode, and the resulting high NO2 photolysis rate provided sufficient conditions for the generation of O3, especially in the upper and middle layers of the boundary layer (Dickerson, 1997; He et al., 2021). Compared to the clean period, lower RH, higher PBLH, predominantly weak northerly winds at the surface (negative  $V_{10}$ ) and stronger downdraft (positive Omega) were found in the first two periods. Lower RH tends to be unfavorable for the wet deposition of O3 (He et al., 2017; Han et al., 2019; Li et al., 2021). The relatively high thermal PBLH allows for adequate mixing of O3 (Li et al., 2018; Zhao et al., 2019; Dong et al., 2020), which is more conducive to the downward transport of O3 in the upper layer when superimposed with the stronger downdraft under the background of the subtropical high and typhoon (Li et al., 2022; Liu et al., 2022). The weak northerly winds favor the transport of high concentrations of locally generated O3 southwards to the coastal areas, resulting in higher O3 concentrations in the south of PRD than in the north during the pollution periods. At the same time, the stable weather under the background of the subtropical high and Typhoon Mina are more favorable for the formation of a deep residual layer, which can store photochemically generated O3 in the daytime and exacerbate the surface O3 concentration through vertical transport in the next day. Furthermore, since the virtual potential temperature  $(\theta_v)$  can represent the height of the atmospheric mixed layer, it can also be seen from the  $\theta_v$  profile results of PRD from September 25 to October 4 at 14:00 in Figure 6c that the inflection point of the  $\theta_v$  in the pollution periods are above 1500 m, that is, the mixed layers are higher, indicating that the relatively higher mixed layer height is more conducive to the mixing of O3.

On the other hand, when PRD was under the influence of Typhoon Mina, it had a higher  $T_2$ , lower RH, a switch to weak northwesterly winds and stronger Omega compared to the subtropical high period, indicating that the more severe meteorological conditions combined with O3 and its precursors accumulated in the subtropical high period were beneficial to the further enhancement of O3 photochemical generation. Further, the key meteorological parameters affecting the changes in O3 concentration in PRD varied as Typhoon Mina moved away from PRD can be seen in Figure 6b:  $T_2$ ,  $W_{10}$ , weak northerly wind (negative  $V_{10}$ ) and Omega all rose first and then gradually decreased after October 1, indicating that the meteorological conditions were more favorable for O3 generation and accumulation when PRD was the influence of typhoon peripheral circulation.

The findings above suggest that the meteorological factors such as lower RH, predominantly weak northerly winds, stronger downdrafts and higher PBLH caused by the subtropical high and Typhoon Mina were the main reasons for the development of this O3 pollution episode, and it ended due to the switch to clean southerly winds at lower levels.

**page 11, line 330 to 333:** Again, the dominant terms are those of CHEM, the value of CHEM is remarkably high in the range of 6.56 to 8.01 ppb/h during the period influenced by the subtropical high and Typhoon Mina (September 25 to October 2, 2019) due to strong photochemical reaction rates resulting from higher SWDOWN and lower RH, but only about 2.84 ppb/h during the clean period of October 3-4 due to the switch to clean southerly winds.

**Specific comments**

1. Line 114: Is the  $O_3$  concentration corresponding to the simulated  $O_3$  in the lowest layer (i.e., below 35 m)? If take the lowest 3 layers into account, especially for periods strongly influenced by downdraft, what would the comparison between the model simulation and the observation look like?

**Response:**

Yes, the simulated  $O_3$  in Figure 5 was obtained from the model's lowest layer, and we have explained it in the revised manuscript (page 6, line 183). As shown in Figure R1 and Table R1, when the lowest 3 layers are combined into one, the comparison between the model simulation and the observation would be closer and the evaluation results would be better. Combined with the daytime vertical contributions of individual processes to  $O_3$  in Figure 11a, it could be seen that since the photochemical generation of  $O_3$  was more intense above the near-surface layer, considering the results of the lowest 3 layers would reduce the negative deviation between the simulation.

Figure R1. Hourly variations of O3 in observed values (blue dots) compared to model simulations of the lowest layer (solid black line) and lowest 3 layers (solid red line) from September 21 to October 4, 2019.

Table R1. Statistical metrics of  $O_3$  in the comparison between the observations and simulations (from lowest layer and lowest 3 layers) from September 21 to October 4,

2019.

|                                            | obs  | sim  | R    | NMB (%) | RMSE  | IOA  |
|--------------------------------------------|------|------|------|---------|-------|------|
| O 3 (ppb, from lowest layer)    | 57.3 | 49.1 | 0.79 | -14.25  | 20.36 | 0.87 |
| O 3 (ppb, from lowest 3 layers) | 57.3 | 52.2 | 0.86 | -8.66   | 15.85 | 0.92 |

2. Line 115: Please define "NAWO" and "CNMC". Is CNMC the same as NEMC in Line 76? If so, please keep the abbreviation consistent.

**Response:**

Thanks for pointing out our negligence. We have described the sources of  $O_3$  data (line 77) and meteorological data (line 84) in section 2.1 of the revised manuscript with new abbreviations. At the same time, the sentence in line 117 is modified by removing the two abbreviations ("NAWO" and "CNMC") for simplicity.

3. Line 133: Please add description on "the second standard of air quality".

**Response:**

Thanks for pointing out our negligence. For hourly  $O_3$  concentration, the national ambient air quality secondary standard is 200 µg m-3 (approximately 93 ppb). We have described it in line 42 of the revised manuscript.

4. Line 137 - 138: Please define "Lev 3" and "Lev 5".

**Response:**

Thanks again for pointing out our negligence. We have added the descriptions of "Lev 1-5" in line 137-139 of the revised manuscript.

5. Line 173: The model overestimated WS10 quite a lot. Which of the wind vector (i.e.,

u, v,  $\omega$ ) has not been well reproduced that could lead to the WS10 overestimation? How would this overestimation further influence the evaluation of the contribution of transport / sea breeze?

**Response:**

The model does overestimate WS10 quite a lot and we acknowledge this referee's concerns that the overestimated model WS10 might further influence the subsequent analysis. For mesoscale models such as WRF, the simulation of the near-surface wind fields under stable weather background is still a challenge (Lim et al., 2018; Srivastava et al., 2021), with most of the previous simulations of near-surface wind speeds being on the high side (Tymvios et al., 2018; Lorenz et al., 2022). We evaluated the variables  $U_{10}$  and  $V_{10}$  which were directly related to  $WS_{10}$  to explore whether the overestimation of model WS10 would affect the subsequent analysis of pollution processes. As can be seen from Figure R2 and Table R2, although the simulation of U10 also has a certain negative deviation with a mean bias (MB) of -0.26m/s, the main reason for the overestimation of WS10 is the larger negative deviation of the  $V_{10}$  simulation (MB of -1.41m/s). That is, the northerly wind simulation is too large, which may overestimate the contribution of the prevailing winds in transporting locally generated O3 in PRD to the sea during the influence of the subtropical high and Typhoon Mina. Although the positive  $V_{10}$  simulations on October 3-5 are large and may overestimate the contribution of clean sea breeze to this O3 episode, the model well simulates the nighttime sea breeze during the pollution periods, providing certain degree of reliability for subsequent studies of the effect of sea breeze on O3. Future studies are planned to improve the simulation of near-surface wind fields by using more detailed terrain data and more suitable parameterization schemes.

Figure R2. Hourly variations of  $U_{10}$  and  $V_{10}$  in observed values (red dots) compared to model simulations (solid lines) during September 21 to October 4, 2019.

Table R2. Statistical metrics of  $U_{10}$  and  $V_{10}$  in the comparison between the observations and simulations during September 21 to October 4, 2019.

|                       | obs  | sim  | R    | MB    | RMSE | ΙΟΑ  |
|-----------------------|------|------|------|-------|------|------|
| U 10 (m/s) | 0.1  | -0.2 | 0.67 | -0.26 | 1.30 | 0.66 |
| V 10 (m/s) | -0.5 | -2.0 | 0.84 | -1.41 | 2.23 | 0.73 |

6. Line 189: Please define  $U_{10}$ ,  $V_{10}$ .

**Response:**

Thanks. We have defined Zonal wind speed at  $10m (U_{10})$  and Meridional wind speed at  $10m (V_{10})$  in line 193 of the revised manuscript.

7. Line 210: Please define  $\theta_v$ .

**Response:**

Thanks. We have added the definition of virtual potential temperature  $(\theta_v)$  in line 212 of the revised manuscript.

8. Line 213 - 216: Higher PBLH could result in higher  $O_3$  concentration due to enhanced contribution of downward  $O_3$  transport. However, higher PBLH could also favor the dilution of  $O_3$  and its precursors, thus result in weaker  $O_3$  production and accumulation. What would be the balance between these two effects?

Response: The referee made a very insightful point that higher PBLH may facilitate downward transport of upper O3, but may also favor the dilution of O3 and its precursors. However, for the thermal boundary layer affected by surface thermodynamics, the stronger the solar radiation or the higher the air temperature, the higher the PBLH, and the meteorological conditions at that time tend to favor the photochemistry of ozone. Previous studies have shown that high PBLH was often accompanied by high concentrations of O3 (Zhao et al., 2019; Dong et al, 2020; He et al., 2021; Liu et al., 2022). In this study, as can be seen from the PBLH heights for each period in Figure 6a and the results from Table 3, the PBLH was higher during the two pollution periods (September 25-28, September 29 to October 2), and the O3 concentration within the boundary layer was also higher during these periods, with 81.53 ppb and 83.04 ppb, respectively. In these cases, the VTRA were -1.83 ppb/h and -0.05 ppb/h, respectively, indicating that higher O3 concentrations within the PBL resulted in vertically upward transport of O3 to the upper atmosphere. In contrast, the daytime PBLH on October 3-4 was lower and a VTRA of 1.80 ppb/h, i.e. O3 above the PBL was transported downward. Meanwhile, photochemical production of O3 was stronger when the boundary layer was higher during the two pollution periods, with CHEM of 8.01 ppb/h and 6.56 ppb/h, respectively, but was only 2.84 ppb/h in during the clean period (October 3-4), suggesting a negligible dilution effect. In summary, under the influence of a subtropical high and Typhoon Mina, high PBLH and associated meteorological conditions were highly conducive to the photochemical production of O3 in the boundary layer in PRD, which resulted in extremely high O3 concentrations within the PBL and a net vertical O3 transport to the upper atmosphere.

9. Figure 5: It looks like there is a 1-hour time shift between the simulated and the observed  $O_3$  concentration. This could be caused by the definition of the measurement time of the CNEC  $O_3$  data (i.e., data at 1:00 represent the averages in 0:00 – 1:00). If this time shift has been taken into account, would the discrepancy between the simulation and the observation become smaller?

**Response:**

We are grateful for the referee's thoughtful point on "the 1-hour time shift between the simulated and the observed O3 concentration could be caused by the definition of the measurement time of the observed O3 data". We looked through the Specifications and Test Procedures for Ambient Air Quality Continuous Automated Monitoring System for SO2, NO2, O3 and CO (HJ 654-2013) developed by China National Environmental Monitoring Center (CNEMC). The O3 data reported at a certain hour was indeed the average of measurements for the hour before (e.g. data at 1:00 represent the average values in 0:01 - 1:00). In comparison, the Operational Guidance for the Community Multiscale Air Quality (CMAQ) Modeling System CMAQ Version 4.7.1 shows that "the 2-D CCTM integral average concentration file (ACONC) contains average model species concentrations for each model hour, as opposed to instantaneous concentrations at the end of each output time step", and line 174 in the source code wr aconc.F,v for calculating ACONC under the path of */cmaq4.7.1/models/CCTM/src/driver/ctm/* shows that "Timestamp represents beginning computed date/time". That is, the ACONC O3 concentration at a certain hour used in the simulation evaluation in this paper was the average of the model values for the hour after the reported time. Therefore, there is indeed a 1-hour time shift between the simulated and the observed O3 concentration. As can be seen from Figure R3 and Table R3, if this time shift is taken into account, the discrepancy between the simulation and the observation does become smaller, with R reaching

0.89. We have corrected the corresponding parts of the figures (Figure 5 and Figure 7) and evaluation (Table 2) that using the ACONC data in the revised manuscript.